# A FASTER PARAMETER-FREE REGRET MATCHING ALGORITHM

**Linjian Meng**[1, 2]**, Youzhi Zhang**[3†]**, Shangdong Yang**[4]**, Wenbin Li**[2]**, Tianyu Ding**[5]**, Yang Gao**[2†]

[1] Shanghai Artificial Intelligence Laboratory
[2] National Key Laboratory for Novel Software Technology, Nanjing University
[3] Centre for Artificial Intelligence and Robotics, Hong Kong Institute of Science & Innovation, CAS
[4] Jiangsu Key Laboratory of Big Data Security and Intelligent Processing, Nanjing
  University of Posts and Telecommunications
[5] Microsoft Corporation
menglinjian@smail.nju.edu.cn, youzhi.zhang@cair-cas.org.hk, sdyang@njupt.edu.cn,
liwenbin@nju.edu.cn tianyuding@microsoft.com, gaoy@nju.edu.cn

## ABSTRACT

Regret Matching (RM) and its variants are widely employed to learn a Nash equilibrium (NE) in large-scale games. However, most existing research only establishes a theoretical convergence rate of $O(1/\sqrt{T})$ for these algorithms in learning an NE. Recent studies have shown that smooth $RM^+$ variants, the advanced variants of RM, can achieve an improved convergence rate of $O(1/T)$. Despite this improvement, smooth $RM^+$ variants lose the parameter-free property, i.e., no parameters that need to be tuned, a highly desirable feature in practical applications. In this paper, we propose a novel smooth $RM^+$ variant called Monotone Increasing Smooth Predictive Regret Matching$^+$ (MI-SPRM$^+$), which retains the parameter-free property while still achieving a theoretical convergence rate of $O(1/T)$. To achieve these properties, MI-SPRM$^+$ employs a technology called Adaptive Regret Domain (ARD), which ensures that the lower bound for the 1-norm of accumulated regrets increases monotonically by adjusting the decision space at each iteration. This design is motivated by the observation that the range of step sizes supporting the $O(1/T)$ convergence rate in existing smooth $RM^+$ variants is contingent on the lower bound for the 1-norm of accumulated regrets. Experimental results confirm that MI-SPRM$^+$ empirically attains an $O(1/T)$ convergence rate.

## 1 INTRODUCTION

Game theory serves as a powerful framework for modeling interactions among multiple agents. A widely studied solution concept in this context is the Nash equilibrium (NE), a state where no player can achieve a higher payoff by unilaterally deviating from their current strategy. This concept provides insight into the stability of decisions, as no player has an incentive to deviate from their chosen strategy once equilibrium is reached. NE is widely applicable across various fields such as economics, political science, business, and international relations, offering a theoretical framework for predicting behavior and optimizing decision-making in complex systems (Osborne, 1994).

To learn an NE in real-world games, compared to traditional NE learning algorithms like Fictitious Play (Brown, 1951), the algorithms based on the regret minimization framework (Zinkevich, 2003), also called regret minimization algorithms, are more popular, particularly given the recent breakthroughs in superhuman game AIs, which rely heavily on this framework (Bowling et al., 2015; Moravčík et al., 2017; Brown and Sandholm, 2018; 2019b; Pérolat et al., 2022). Among regret minimization algorithms, Regret Matching (RM) (Hart and Mas-Colell, 2000) and its variants (Gordon, 2006; Lanctot et al., 2009; Johanson et al., 2012; Lanctot, 2013; Tammelin, 2014;

---

† Corresponding authors.

Brown and Sandholm, 2019a; Farina et al., 2021; Zhang et al., 2022; Xu et al., 2022; Farina et al., 2023; Xu et al., 2024b) stand out in practical applications due to their superior empirical performance compared to other regret minimization algorithms, such as those based on Online Mirror Descent (OMD) (Nemirovskij and Yudin, 1983) or Follow the Regularized Leader (FTRL) (Shalev-Shwartz and Singer, 2007). For example, RM variants are widely used in superhuman Poker AIs (Bowling et al., 2015; Moravčík et al., 2017; Brown and Sandholm, 2018; 2019b), while OMD/FTRL-based algorithms are not.

Numerous studies have shown that many advanced algorithms based on OMD and FTRL achieve an $O(1/T)$ theoretical convergence rate (Rakhlin and Sridharan, 2013a;b; Syrgkanis et al., 2015; Farina et al., 2019; Piliouras et al., 2022; Hsieh et al., 2021). Unfortunately, although several advanced RM variants have been proposed, such as Regret Matching$^+$ (RM$^+$) (Tammelin, 2014), Discounted Regret Matching (DRM) (Brown and Sandholm, 2019a), and Predictive Regret Matching$^+$ (PRM$^+$) (Farina et al., 2021), they only achieve an $O(1/\sqrt{T})$ theoretical convergence rate.

To address the problem that RM variants only achieve an $O(1/\sqrt{T})$ theoretical convergence rate, smooth RM$^+$ variants like Smooth Predictive Regret Matching$^+$ (SPRM$^+$) (Farina et al., 2023) are introduced. These variants achieve an $O(1/T)$ theoretical convergence rate by ensuring that the lower bound for the 1-norm of accumulated regrets consistently exceeds a positive constant. However, smooth RM$^+$ variants lose the parameter-free property of other RM$^+$ variants like RM$^+$ and PRM$^+$, i.e., *no parameters need to be tuned* (Grand-Clément and Kroer, 2021), which is a highly desirable feature in practical applications. Specifically, one of the most significant obstacles in using regret minimization algorithms to learn an NE in games is the sensitivity to hyperparameters. As illustrated in our experiments (Section 4), even in the game considered in the original paper of SPRM$^+$ (Farina et al., 2023), the convergence rate of SPRM$^+$ varies significantly depending on the choice of step size $\eta$[1]. Specifically, after $10^5$ iterations, SPRM$^+$ ($\eta = 0.1$) achieves a duality gap (the distance to NE; lower is better) that is 10 and 5 times smaller than SPRM$^+$ ($\eta = 0.01$) and SPRM$^+$ ($\eta = 1$), respectively. This sensitivity implies that extensive parameter tuning is required to identify a suitable parameter. In contrast, parameter-free algorithms eliminate the need for such tuning, allowing the algorithm to directly learn an NE without any tuning of parameters.

To recover the parameter-free property[2] for smooth RM$^+$ variants, we propose a novel smooth RM$^+$ variant called *Monotone Increasing Smooth Predictive Regret Matching$^+$* (MI-SPRM$^+$), a parameter-free algorithm that achieves an $O(1/T)$ theoretical convergence rate. The key insight of MI-SPRM$^+$, which enables the simultaneous achievement of the parameter-free property and an $O(1/T)$ theoretical convergence rate, is that the appropriate range of step sizes for achieving this rate in SPRM$^+$ depends on the lower bound for the 1-norm of accumulated regrets. Therefore, MI-SPRM$^+$ employs a technology called *Adaptive Regret Domain* (ARD), which ensures this lower bound monotonically increases by adjusting the decision space at each iteration to achieve an $O(1/T)$ theoretical convergence rate with the parameter-free property.

Furthermore, we evaluate the empirical convergence rate of MI-SPRM$^+$ on the games considered in the original paper of smooth RM$^+$ variants (Farina et al., 2023) and randomly generated two-player zero-sum NFGs. The experimental results show that MI-SPRM$^+$ consistently attains an $O(1/T)$ empirical convergence rate across all evaluated games. More interestingly, MI-SPRM$^+$ outperforms all other tested algorithms, including existing RM variants and traditional non-parameter-free and parameter-free regret minimization algorithms.

## 2  RELATED WORK

**Traditional parameter-free regret minimization algorithms.** Although many results about the $O(1/T)$ theoretical convergence rate of traditional regret minimization algorithms based on OMD

---

[1]Traditional regret minimization algorithms are even more sensitive to parameters than SPRM$^+$ as they will diverge without suitable parameters.

[2]Beyond being parameter-free, RM$^+$ and PRM$^+$ exhibits a property called stepsize-invariance (Chakrabarti et al., 2024), meaning that the algorithm's output remains unchanged regardless of the parameter choice. This property is also referred to as strongly parameter-free (Grand-Clément and Kroer, 2021). See details in Section 5. In this paper, we only focus on the parameter-free property since to the best of our knowledge, no algorithm simultaneously achieves both an $O(1/T)$ theoretical convergence rate and stepsize-invariance.

or FTRL have been proposed, these algorithms are highly sensitive to the choice of parameters. Specifically, as demonstrated in our experiments, these algorithms exhibit an empirical convergence rate of $O(1/T)$ with appropriate parameter tuning, while they either converge very slowly or may even diverge when the parameters are poorly chosen. To establish the parameter-free regret minimization algorithms, the most common method is the *doubling trick* (Auer et al., 1995). However, the doubling trick can only create parameter-free regret minimization algorithms with an $O(\log_2 T/T)$ convergence rate even if the original regret minimization algorithm theoretically guarantees an $O(1/T)$ convergence rate. Specifically, consider that the original regret minimization algorithm exhibits a regret bound of $C$ for any number of iterations $T$, thus implying an $O(1/T)$ convergence rate. For a total of $T$ iterations where there exists a positive constant $M$ such that $2^M \leq T < 2^{M+1}$, the cumulative regret resulting from the application of the doubling trick is bounded by $\sum_{m=1}^{M+1} C$. This summation results in a regret bound of $O(\log_2 T)$, leading to a convergence rate of $O(\log_2 T/T)$. To achieve both the parameter-free property and a theoretical convergence rate of $O(1/T)$, Hsieh et al. (2021) propose an algorithm called *Dual Stabilized Optimistic Mirror Descent* (DS-OptMD), which achieves this goal by autonomously learning the step size. However, our experimental results reveal that the empirical convergence rate of DS-OptMD is significantly slower than $O(1/T)$.

**Regret Matching (RM) variants.** The key distinction between RM variants and traditional regret minimization algorithms based on OMD or FTRL is that RM variants update within the (subset of the) non-negative orthant, whereas the latter update within the original strategy space of the game (Farina et al., 2021; 2023). These algorithms usually perform numerically better than traditional regret minimization algorithms. However, although many technologies have been proposed to improve these algorithms' empirical convergence rate, these algorithms only achieve an $O(1/\sqrt{T})$ theoretical convergence rate. To provide an $O(1/T)$ convergence rate to $RM^+$ variants, Farina et al. (2023) propose smooth $RM^+$ variants like *Smooth Predictive $RM^+$* (SPRM$^+$). Although smooth $RM^+$ variants achieve an $O(1/T)$ theoretical convergence rate, they drop a very appealing property in most other $RM^+$ variants—the parameter-free property, which is extremely useful in practice as it avoids fine tuning the parameters.

To the best of our knowledge, we introduce the first parameter-free RM variant that achieves an $O(1/T)$ theoretical convergence rate, named *Monotone Increasing Smooth Predictive Regret Matching$^+$* (MI-SPRM$^+$). To achieve both the parameter-free property and a theoretical convergence rate of $O(1/T)$, MI-SPRM$^+$ employs ARD, dynamically adjusting the decision space at each iteration, while SPRM$^+$ maintains a fixed decision space throughout each iteration (a detailed comparison is shown in Section 4). In addition, unlike DS-OptMD, which theoretically has an $O(1/T)$ convergence rate but empirically demonstrates a much slower convergence rate, MI-SPRM$^+$ achieves an $O(1/T)$ convergence rate both in theory and in practice.

## 3 PRELIMINARIES

**Two-player zero-sum normal-form games (NFGs).** In this paper, we study two-player zero-sum NFGs, which encompass many classic scenarios like Rock-Paper-Scissors. In these games, each player $i \in \mathcal{N} = \{0, 1\}$ simultaneously selects an action $a_i \in \mathcal{A}_i$ and receives a reward $r_i(a_i, a_{1-i})$, where $\mathcal{A}_i$ denotes the action space of player $i$ and $\mathcal{N}$ is the set of players. We denote the strategy of player $i$ by $\boldsymbol{x}_i$, which is the probability distribution over all actions $a_i \in \mathcal{A}_i$. The set of strategies is denoted as $\boldsymbol{\mathcal{X}}_i$, which is a $(|\mathcal{A}_i| - 1)$-dimensional simplex, *i.e.*, $\forall \boldsymbol{x}_i \in \boldsymbol{\mathcal{X}}_i$, $\boldsymbol{x}_i \geq \boldsymbol{0}$ and $\langle \boldsymbol{1}, \boldsymbol{x}_i \rangle = 1$, implying $\|\boldsymbol{x}_i\|_2 \leq \|\boldsymbol{x}_i\|_1 = 1$. Similarly, the strategy profile is represented by $\boldsymbol{x} = [\boldsymbol{x}_0; \boldsymbol{x}_1]$, and the set of strategy profiles is denoted as $\boldsymbol{\mathcal{X}} = \boldsymbol{\mathcal{X}}_0 \times \boldsymbol{\mathcal{X}}_1$. The set $\boldsymbol{\mathcal{X}}$ is a compact set because each $\boldsymbol{\mathcal{X}}_i$ is a simplex, which is a compact set. The utility of player $i$ when all players follow the strategy profile $\boldsymbol{x}$ is given by $u_i(\boldsymbol{x}) = u_i(\boldsymbol{x}_i, \boldsymbol{x}_{1-i}) = \sum_{a_0 \in \mathcal{A}_0} \sum_{a_1 \in \mathcal{A}_1} r_i(a_0, a_1) \boldsymbol{x}_0(a_0) \boldsymbol{x}_1(a_1)$. The zero-sum property implies that $\sum_{i \in \mathcal{N}} u_i(\boldsymbol{x}) = 0$. The loss gradient for player $i$ is denoted by $\boldsymbol{\ell}_i^{\boldsymbol{x}} = -\nabla_{\boldsymbol{x}_i} u_i(\boldsymbol{x}_i, \boldsymbol{x}_{1-i})$. We assume that $\forall \boldsymbol{x}, \boldsymbol{x}' \in \boldsymbol{\mathcal{X}}$,

$$\|\boldsymbol{\ell}^{\boldsymbol{x}} - \boldsymbol{\ell}^{\boldsymbol{x}'}\|_2 \leq L\|\boldsymbol{x} - \boldsymbol{x}'\|_2, \ \|\boldsymbol{\ell}^{\boldsymbol{x}}\|_1 \leq P, \tag{1}$$

where $\boldsymbol{\ell}^{\boldsymbol{x}} = [\boldsymbol{\ell}_0^{\boldsymbol{x}}; \boldsymbol{\ell}_1^{\boldsymbol{x}}]$, and $L, P > 0$ are constants. The assumptions in Eq. (1) are among the most fundamental in game solving (Farina et al., 2023; Cai and Zheng, 2023; Cai et al., 2024; 2025). We also use $D$ to denote $\max_{i \in \mathcal{N}} |\mathcal{A}_i|$.

**Nash equilibrium (NE).** To solve two-player zero-sum NFGs, a common goal is the NE where no player can benefit from deviating unilaterally from this equilibrium. In other words, for any player, her strategy is the best-response to the strategies of others. We use $\mathcal{X}^*$ to denote the set of NE. As analyzed in Facchinei (2003), if $\boldsymbol{x}^* \in \mathcal{X}^*$, then $\langle \boldsymbol{\ell}_i^{\boldsymbol{x}^*}, \boldsymbol{x}_i^* - \boldsymbol{x}_i \rangle \leq 0, \forall \boldsymbol{x}_i \in \mathcal{X}_i$. We use the duality gap as the metric to measure the distance from strategy profile $\boldsymbol{x}$ to $\mathcal{X}^*$. Precisely, the duality gap of strategy profile $\boldsymbol{x}$ is defined as $dg(\boldsymbol{x}) = \max_{\boldsymbol{x}' \in \mathcal{X}} \langle \boldsymbol{\ell}^{\boldsymbol{x}}, \boldsymbol{x} - \boldsymbol{x}' \rangle = \sum_{i \in \mathcal{N}} \max_{\boldsymbol{x}_i \in \mathcal{X}_i} \langle \boldsymbol{\ell}_i^t, \boldsymbol{x}_i^t - \boldsymbol{x}_i \rangle$. If $dg(\boldsymbol{x}) \leq \delta$, then $\boldsymbol{x}$ is a $\delta$-approximate NE ($\delta$-NE). If and only if $\boldsymbol{x}$ is a NE, $dg(\boldsymbol{x}) = 0$.

**Regret minimization framework.** To learn an NE in a two-player zero-sum NFG, the most popular algorithms are the algorithms based on the regret minimization framework (Zinkevich, 2003), also called regret minimization algorithms (Zinkevich, 2003; Hart and Mas-Colell, 2000; Nemirovskij and Yudin, 1983; Rakhlin and Sridharan, 2013b; Syrgkanis et al., 2015; Farina et al., 2019; Piliouras et al., 2022; Hsieh et al., 2021; Gordon, 2006; Lanctot et al., 2009; Johanson et al., 2012; Lanctot, 2013; Tammelin, 2014; Brown and Sandholm, 2019a; Farina et al., 2021; Zhang et al., 2022; Xu et al., 2022; Farina et al., 2023; Xu et al., 2024b). In this framework, each player $i$ selects a decision $\boldsymbol{x}_i^t \in \mathcal{X}_i$ according to feedback received from the game. In games solving, such feedback is set to the loss gradient $\boldsymbol{\ell}_i^{t-1} = \boldsymbol{\ell}_i^{\boldsymbol{x}^{t-1}}$. The goal of regret minimization algorithms is to enable the regret $\sum_{t=1}^{T} \langle \boldsymbol{\ell}_i^t, \boldsymbol{x}_i^t - \boldsymbol{x}_i \rangle, \forall \boldsymbol{x}_i \in \mathcal{X}_i$ to grow sublinearly. After $T$ iterations, let $\boldsymbol{\ell}^t = [\boldsymbol{\ell}_0^t; \boldsymbol{\ell}_1^t]$, $\boldsymbol{x}^t = [\boldsymbol{x}_0^t; \boldsymbol{x}_1^t]$, and $\boldsymbol{x} = [\boldsymbol{x}_0; \boldsymbol{x}_1]$, suppose the social regret $\sum_{t=1}^{T} \langle \boldsymbol{\ell}^t, \boldsymbol{x}^t - \boldsymbol{x} \rangle = \sum_{i \in \mathcal{N}} \sum_{t=1}^{T} \langle \boldsymbol{\ell}_i^t, \boldsymbol{x}_i^t - \boldsymbol{x}_i \rangle$ satisfies $\sum_{t=1}^{T} \langle \boldsymbol{\ell}^t, \boldsymbol{x}^t - \boldsymbol{x} \rangle / T \leq \varepsilon, \forall \boldsymbol{x} \in \mathcal{X}$. Then, the time-averaged strategy profile $\bar{\boldsymbol{x}}^T = \sum_{t=1}^{T} \boldsymbol{x}^t / T$ is an $\varepsilon$-NE. The $O(1/T)$ theoretical convergence rate of the regret minimization algorithms is that $\sum_{t=1}^{T} \langle \boldsymbol{\ell}^t, \boldsymbol{x}^t - \boldsymbol{x} \rangle / T \leq O(1/T)$ after $T \geq 1$ iterations.

**Online Mirror Descent (OMD).** Among regret minimization algorithms, one of the most classic is OMD (Nemirovskij and Yudin, 1983). Let $\phi(\cdot) : \mathcal{X}_i \to \mathbb{R}$, OMD generates decisions via

$$\boldsymbol{x}_i^{t+1} \in \arg\min_{\boldsymbol{x}_i \in \mathcal{X}_i} \{ \langle \boldsymbol{\ell}_i^t, \boldsymbol{x}_i \rangle + \frac{1}{\eta} \mathcal{B}_\phi(\boldsymbol{x}_i, \boldsymbol{x}_i^t) \}, \tag{2}$$

where the step size $\eta > 0$ is a constant, $\mathcal{B}_\phi(\boldsymbol{a}, \boldsymbol{b}) = \phi(\boldsymbol{a}) - \phi(\boldsymbol{b}) - \langle \nabla \phi(\boldsymbol{b}), \boldsymbol{a} - \boldsymbol{b} \rangle$ is the Bregman divergence associated with $\phi(\cdot)$. In Eq. (2), $\phi(\cdot)$ can be any strongly convex function.

**Regret Matching$^+$ (RM$^+$).** To solve real-world games, RM variants are among the most widely used regret minimization algorithms, as demonstrated by their success in superhuman Poker AIs (Bowling et al., 2015; Moravčík et al., 2017; Brown and Sandholm, 2018; 2019b; Pérolat et al., 2022). In this paper, we focus on RM$^+$ variants (Tammelin, 2014; Farina et al., 2021; 2023; Meng et al., 2023) since RM$^+$ variants usually outperform vanilla RM (Hart and Mas-Colell, 2000; Gordon, 2006; Zinkevich et al., 2007). At each iteration $t \geq 1$, RM$^+$ updates its accumulated regret $\hat{\boldsymbol{\theta}}_i^t$ via the *regret matching$^+$ operator*: $\boldsymbol{\theta}_i^{t+1} = [\boldsymbol{\theta}_i^t + \eta \boldsymbol{F}_i^t(\boldsymbol{x}^t)]^+$, where the step size $\eta > 0$ is a constant, $\boldsymbol{\theta}_i^0 = \boldsymbol{0}$, $\boldsymbol{x}_i^0 = \boldsymbol{1}/|\mathcal{A}_i|$, $\boldsymbol{x}_i^t = \boldsymbol{\theta}_i^t / \|\boldsymbol{\theta}_i^t\|_1$ ($t \geq 1$), $\boldsymbol{F}_i^t(\boldsymbol{x}^t) = \boldsymbol{F}_i^t(\boldsymbol{\theta}^t) = \langle \boldsymbol{\ell}_i^t, \boldsymbol{x}_i^t \rangle \boldsymbol{1} - \boldsymbol{\ell}_i^t$ ($\boldsymbol{\theta}^t = [\boldsymbol{\theta}_0^t; \boldsymbol{\theta}_1^t]$) is the instantaneous regret, and $[\cdot]^+ = \max(\cdot, \boldsymbol{0})$. From the analysis in Farina et al. (2021), RM$^+$ is connected to an OMD instance which performs updates in the non-negative orthant and sets $\phi(\cdot)$ as the quadratic regularizer $\psi(\cdot) = \| \cdot \|_2^2 / 2$. Formally, RM+ can be rewritten as

$$\boldsymbol{\theta}_i^{t+1} \in \arg\min_{\boldsymbol{\theta}_i \in \mathbb{R}_{\geq 0}^{|\mathcal{A}_i|}} \{ \langle -\boldsymbol{F}_i^t(\boldsymbol{\theta}^t), \boldsymbol{\theta}_i \rangle + \frac{1}{\eta} \mathcal{B}_\psi(\boldsymbol{\theta}_i, \boldsymbol{\theta}_i^t) \}, \quad \boldsymbol{x}_i^{t+1} = \frac{\boldsymbol{\theta}_i^{t+1}}{\|\boldsymbol{\theta}_i^{t+1}\|_1}, \tag{3}$$

where the step size $\eta > 0$ is a constant, and $\mathbb{R}_{\geq 0}^d = \{\boldsymbol{y} | \boldsymbol{y} \in \mathbb{R}^d, \boldsymbol{y} \geq \boldsymbol{0}\}$. Notably, $\psi(\cdot)$ is the quadratic regularizer $\| \cdot \|_2^2 / 2$ implying that $\forall \boldsymbol{a}, \boldsymbol{b} \in \mathbb{R}^d, \mathcal{B}_\psi(\boldsymbol{a}, \boldsymbol{b}) = \mathcal{B}_\psi(\boldsymbol{b}, \boldsymbol{a}) = \|\boldsymbol{a} - \boldsymbol{b}\|_2^2 / 2$.

**Predictive Regret Matching$^+$ (PRM$^+$).** To improve the empirical convergence rate of RM$^+$, Farina et al. (2021) propose PRM$^+$, whose key insight is to make a prediction at each iteration $t$. PRM$^+$ uses the feedback at the last iteration $t - 1$ as the prediction at the current iteration $t$. Formally, at each iteration $t \geq 1$, the update rule of PRM$^+$ is

$$\boldsymbol{\theta}_i^t \in \arg\min_{\boldsymbol{\theta}_i \in \mathbb{R}_{\geq 0}^{|\mathcal{A}_i|}} \{ \langle -\boldsymbol{F}_i^{t-1}(\boldsymbol{\theta}^{t-1}), \boldsymbol{\theta}_i \rangle + \frac{1}{\eta} \mathcal{B}_\psi(\boldsymbol{\theta}_i, \hat{\boldsymbol{\theta}}_i^t) \}, \quad \boldsymbol{x}_i^t = \frac{\boldsymbol{\theta}_i^t}{\|\boldsymbol{\theta}_i^t\|_1},$$

$$\hat{\boldsymbol{\theta}}_i^{t+1} \in \arg\min_{\boldsymbol{\theta}_i \in \mathbb{R}_{\geq 0}^{|\mathcal{A}_i|}} \{ \langle -\boldsymbol{F}_i^t(\boldsymbol{\theta}^t), \boldsymbol{\theta}_i \rangle + \frac{1}{\eta} \mathcal{B}_\psi(\boldsymbol{\theta}_i, \hat{\boldsymbol{\theta}}_i^t) \}, \tag{4}$$

where the step size $\eta > 0$ is a constant, $\boldsymbol{\theta}_i^0 = \mathbf{1}/|A_i|$, $\hat{\boldsymbol{\theta}}_i^1 = \mathbf{0}$, $\boldsymbol{x}_i^0 = \mathbf{1}/|A_i|$, $\boldsymbol{F}_i^{t-1}(\boldsymbol{\theta}^{t-1}) = \langle \boldsymbol{\ell}_i^{t-1}, \boldsymbol{x}_i^{t-1} \rangle \mathbf{1} - \boldsymbol{\ell}_i^{t-1}$, $\boldsymbol{F}_i^t(\boldsymbol{\theta}^t) = \langle \boldsymbol{\ell}_i^t, \boldsymbol{x}_i^t \rangle \mathbf{1} - \boldsymbol{\ell}_i^t$, $\boldsymbol{\theta}^{t-1} = [\boldsymbol{\theta}_0^{t-1}; \boldsymbol{\theta}_1^t]$, $\boldsymbol{\theta}^t = [\boldsymbol{\theta}_0^t; \boldsymbol{\theta}_1^t]$, and $\psi(\cdot)$ is the quadratic regularizer defined in Eq. (3). In Eq. (4), $\boldsymbol{F}_i^{t-1}(\boldsymbol{\theta}^{t-1})$ is the prediction at iteration $t$. Note that the term $\langle \boldsymbol{\ell}_i^t, \boldsymbol{x}_i^t \rangle \mathbf{1}$ ($\langle \boldsymbol{\ell}_i^{t-1}, \boldsymbol{x}_i^{t-1} \rangle \mathbf{1}$) in $\boldsymbol{F}_i^t(\boldsymbol{\theta}^t)$ ($\boldsymbol{F}_i^{t-1}(\boldsymbol{\theta}^{t-1})$) is a $|\mathcal{A}_i|$-dimensional vector as $\boldsymbol{\ell}_i^t$ ($\boldsymbol{\ell}_i^{t-1}$) in $\boldsymbol{F}_i^t(\boldsymbol{\theta}^t)$ ($\boldsymbol{F}_i^{t-1}(\boldsymbol{\theta}^{t-1})$) is a $|\mathcal{A}_i|$-dimensional vector (from the definition of $\boldsymbol{\ell}_i^t$, the shape of $\boldsymbol{\ell}_i^t$ is consistent with that of $\boldsymbol{x}_i$). As analyzed in Farina et al. (2021), if $\|\boldsymbol{F}^{t-1}(\boldsymbol{\theta}^{t-1}) - \boldsymbol{F}^t(\boldsymbol{\theta}^t)\|_2 = 0$ ($\boldsymbol{F}^{t-1}(\boldsymbol{\theta}^{t-1}) = [\boldsymbol{F}_0^{t-1}(\boldsymbol{\theta}^{t-1}); \boldsymbol{F}_1^{t-1}(\boldsymbol{\theta}^{t-1})]$ and $\boldsymbol{F}^t(\boldsymbol{\theta}^t) = [\boldsymbol{F}_0^t(\boldsymbol{\theta}^t); \boldsymbol{F}_1^t(\boldsymbol{\theta}^t)]$) holds, PRM$^+$ guarantees that its average strategy profile, $\bar{\boldsymbol{x}}^T = \sum_{t=1}^T \boldsymbol{x}^t/T$, converges to an approximate NE with a theoretical convergence rate of $O(1/T)$, where $\boldsymbol{x}^t = [\boldsymbol{x}_0^t; \boldsymbol{x}_1^t]$. However, due to the instability (Farina et al., 2023), *i.e.*, rapid fluctuations of the instantaneous regret $\boldsymbol{F}_i^t(\boldsymbol{\theta}^t)$ across iterations, this assumption does not hold. Therefore, PRM$^+$ only achieves an $O(1/\sqrt{T})$ theoretical convergence rate. Notably, RM$^+$ and PRM$^+$ are parameter-free algorithms, as the sequence of strategy profiles they generate remains invariant under any choice of $\eta$ (Farina et al., 2021; 2023). For further details, refer to Section 5.

**Smooth Regret Matching$^+$ variants (Farina et al., 2023).** Smooth RM$^+$ variants are designed to address the instability of PRM$^+$ and obtain an $O(1/T)$ theoretical convergence rate. Our algorithms are based on *Smooth Predictive Regret Matching$^+$* (SPRM$^+$) (Farina et al., 2023) as SPRM$^+$ is a single-call algorithm, which only calls loss gradients once at each iteration, while other smooth RM$^+$ variants are not. To address the instability of PRM$^+$ and achieve an $O(1/T)$ theoretical convergence rate, SPRM$^+$ performs updates in the space $\mathbb{R}_{\geq R}^{|\mathcal{A}_i|}$, whereas PRM$^+$ performs updates in the space $\mathbb{R}_{\geq 0}^{|\mathcal{A}_i|}$. This modification ensures $\|\boldsymbol{F}^{t-1}(\boldsymbol{\theta}^{t-1}) - \boldsymbol{F}^t(\boldsymbol{\theta}^t)\|_2^2 \leq O(\|\boldsymbol{\theta}^{t-1} - \boldsymbol{\theta}^t\|_2^2)$ to reduce the instability. More precisely, Farina et al. (2023) employ the property that $\|\boldsymbol{F}^{t-1}(\boldsymbol{\theta}^{t-1}) - \boldsymbol{F}^t(\boldsymbol{\theta}^t)\|_2^2 \leq O(\|\boldsymbol{\theta}^{t-1} - \boldsymbol{\theta}^t\|_2^2)$ to prove the $O(1/T)$ theoretical convergence rate. See Appendix B for the details of the proof. Formally, the update rule of SPRM$^+$ at iteration $t \geq 1$ is

$$\boldsymbol{\theta}_i^t \in \underset{\boldsymbol{\theta}_i \in \mathbb{R}_{\geq R}^{|\mathcal{A}_i|}}{\arg\min} \{\langle -\boldsymbol{F}_i^{t-1}(\boldsymbol{\theta}^{t-1}), \boldsymbol{\theta}_i \rangle + \frac{1}{\eta} \mathcal{B}_\psi(\boldsymbol{\theta}_i, \hat{\boldsymbol{\theta}}_i^t)\}, \, \boldsymbol{x}_i^t = \frac{\boldsymbol{\theta}_i^t}{\|\boldsymbol{\theta}_i^t\|_1},$$

$$\hat{\boldsymbol{\theta}}_i^{t+1} \in \underset{\boldsymbol{\theta}_i \in \mathbb{R}_{\geq R}^{|\mathcal{A}_i|}}{\arg\min} \{\langle -\boldsymbol{F}_i^t(\boldsymbol{\theta}^t), \boldsymbol{\theta}_i \rangle + \frac{1}{\eta} \mathcal{B}_\psi(\boldsymbol{\theta}_i, \hat{\boldsymbol{\theta}}_i^t)\},$$

(5)

where $R > 0$ is a constant, the step size $\eta > 0$ is a constant, $\boldsymbol{\theta}_i^0 = \mathbf{1}/|A_i|$, $\hat{\boldsymbol{\theta}}_i^1 = \mathbf{1}/|A_i|$, $\boldsymbol{x}_i^0 = \mathbf{1}/|A_i|$, $\boldsymbol{F}_i^{t-1}(\boldsymbol{\theta}^{t-1}) = \langle \boldsymbol{\ell}_i^{t-1}, \boldsymbol{x}_i^{t-1} \rangle \mathbf{1} - \boldsymbol{\ell}_i^{t-1}$, $\boldsymbol{F}_i^t(\boldsymbol{\theta}^t) = \langle \boldsymbol{\ell}_i^t, \boldsymbol{x}_i^t \rangle \mathbf{1} - \boldsymbol{\ell}_i^t$, $\boldsymbol{\theta}^{t-1} = [\boldsymbol{\theta}_0^{t-1}; \boldsymbol{\theta}_1^{t-1}]$, $\boldsymbol{\theta}^t = [\boldsymbol{\theta}_0^t; \boldsymbol{\theta}_1^t]$, and $\psi(\cdot)$ is the quadratic regularizer defined in Eq. (3).

## 4 OUR ALGORITHM

Although SPRM$^+$ is a powerful algorithm, it is not parameter-free, as it requires the fine-tuning of the parameter $\eta$ to achieve an $O(1/T)$ theoretical convergence rate. This dependency on parameter tuning diminishes its practical appeal. To avoid the parameter tuning, we propose a novel RM$^+$ variant called *Monotone Increasing Smooth Predictive Regret Matching$^+$* (MI-SPRM$^+$), a parameter-free algorithm that achieves an $O(1/T)$ theoretical convergence rate.

MI-SPRM$^+$ is inspired by the convergence results of SPRM$^+$, which achieves an $O(1/T)$ theoretical convergence rate with $0 < \eta < RC_0$, where $C_0 = 1/\sqrt{8D(2L^2 + 4DL^2 + 4DP^2)}$ is a game-dependent constant and $R$ is defined in Eq. (5) (the formal convergence result of SPRM$^+$ is detailed in Theorem B.1). It is evident that for any $\eta > 0$, if $R > \eta/C_0$, SPRM$^+$ guarantees an $O(1/T)$ theoretical convergence rate. To achieve this convergence rate with the parameter-free property, a viable approach is to adaptively increase the value of $R$, the lower bound for the 1-norm of accumulated regrets, so that $R$ exceeds $\eta/C_0$. We call this approach *Adaptive Regret Domain* (ARD).

Existing OMD-based algorithms like DS-OptMD (Hsieh et al., 2021), achieve the parameter-free property and an $O(1/T)$ theoretical convergence rate by adaptively reducing the step size $\eta$. However, the reduction method employed in DS-OptMD is too conservative, thereby resulting in a poor empirical convergence rate, as shown in our experiments. In contrast, ARD exploits the convergence property of SPRM$^+$, which simultaneously depend on the lower bound of the 1-norm of accumulated regrets and the value of $\eta$. Instead of reducing $\eta$, ARD adopts a more aggressive approach for

---

**Algorithm 1** MI-SPRM$^+$

1: Initialize: $R^1 = 1$, $\boldsymbol{\theta}_i^0 \leftarrow \frac{1}{|A_i|}$, $\hat{\boldsymbol{\theta}}_i^1 \leftarrow \frac{1}{|A_i|}$, $\boldsymbol{x}_i^0 \leftarrow \frac{1}{|A_i|}$, $\forall i \in \mathcal{N}$
2: $\boldsymbol{x}^0 = [\boldsymbol{x}_0^0; \boldsymbol{x}_1^0]$
3: **for** $t = 1, 2, \ldots, T$ **do**
4:   **for** $i \in \mathcal{N}$ **do**
5:     $\boldsymbol{\ell}_i^{t-1} = -\nabla_{\boldsymbol{x}_i^{t-1}} u_i(\boldsymbol{x}^{t-1})$, $\boldsymbol{F}_i^{t-1}(\boldsymbol{\theta}^{t-1}) = \langle \boldsymbol{\ell}_i^{t-1}, \boldsymbol{x}_i^{t-1} \rangle \mathbf{1} - \boldsymbol{\ell}_i^{t-1}$
6:     $\boldsymbol{\theta}_i^t \in \arg\min_{\boldsymbol{\theta}_i \in \mathbb{R}_{\geq R^t}^{|\mathcal{A}_i|}} \left\{ \langle -\boldsymbol{F}_i^{t-1}(\boldsymbol{\theta}^{t-1}), \boldsymbol{\theta}_i \rangle + \mathcal{B}_\psi(\boldsymbol{\theta}_i, \hat{\boldsymbol{\theta}}_i^t) \right\}$, $\boldsymbol{x}_i^t = \frac{\boldsymbol{\theta}_i^t}{\|\boldsymbol{\theta}_i^t\|_1}$
7:   **end for**
8:   **for** $i \in \mathcal{N}$ **do**
9:     $\boldsymbol{\ell}_i^t = -\nabla_{\boldsymbol{x}_i^t} u_i(\boldsymbol{x}^t)$, $\boldsymbol{F}_i^t(\boldsymbol{\theta}^t) = \langle \boldsymbol{\ell}_i^t, \boldsymbol{x}_i^t \rangle \mathbf{1} - \boldsymbol{\ell}_i^t$
10:    $\hat{\boldsymbol{\theta}}_i^{t+1} \in \arg\min_{\boldsymbol{\theta}_i \in \mathbb{R}_{\geq R^t}^{|\mathcal{A}_i|}} \left\{ \langle -\boldsymbol{F}_i^t(\boldsymbol{\theta}^t), \boldsymbol{\theta}_i \rangle + \mathcal{B}_\psi(\boldsymbol{\theta}_i, \hat{\boldsymbol{\theta}}_i^t) \right\}$
11:   **end for**
12:   $\boldsymbol{\theta}^{t-1} = [\boldsymbol{\theta}_0^{t-1}; \boldsymbol{\theta}_1^{t-1}]$, $\boldsymbol{\theta}^t = [\boldsymbol{\theta}_0^t; \boldsymbol{\theta}_1^t]$, $\boldsymbol{x}^t = [\boldsymbol{x}_0^t; \boldsymbol{x}_1^t]$
13:   $\boldsymbol{F}^{t-1}(\boldsymbol{\theta}^{t-1}) = [\boldsymbol{F}_0^{t-1}(\boldsymbol{\theta}^{t-1}); \boldsymbol{F}_1^{t-1}(\boldsymbol{\theta}^{t-1})]$, $\boldsymbol{F}^t(\boldsymbol{\theta}^t) = [\boldsymbol{F}_0^t(\boldsymbol{\theta}^t); \boldsymbol{F}_1^t(\boldsymbol{\theta}^t)]$
14:   $R^{t+1} = \begin{cases} R^t + 1 & \text{if } \|F^t(\boldsymbol{\theta}^t) - F^{t-1}(\boldsymbol{\theta}^{t-1})\|_2^2 - \frac{\mathcal{B}_\psi(\hat{\boldsymbol{\theta}}^t, \boldsymbol{\theta}^{t-1}) + \mathcal{B}_\psi(\hat{\boldsymbol{\theta}}^t, \boldsymbol{\theta}^t)}{2} > 0 \\ R^t & \text{else} \end{cases}$,
15: **end for**
16: **return** $\bar{\boldsymbol{x}}^T = \frac{\sum_{t=1}^T R^t \boldsymbol{x}^t}{\sum_{t=1}^T R^t}$

---

increasing the lower bound of the 1-norm of accumulated regrets, maintaining a faster empirical convergence rate. Building on ARD, we propose MI-SPRM$^+$, whose updates follow the recursion:

$$\boldsymbol{\theta}_i^t \in \underset{\boldsymbol{\theta}_i \in \mathbb{R}_{\geq R^t}^{|\mathcal{A}_i|}}{\arg\min} \left\{ \langle -\boldsymbol{F}_i^{t-1}(\boldsymbol{\theta}^{t-1}), \boldsymbol{\theta}_i \rangle + \mathcal{B}_\psi(\boldsymbol{\theta}_i, \hat{\boldsymbol{\theta}}_i^t) \right\}, \boldsymbol{x}_i^t = \frac{\boldsymbol{\theta}_i^t}{\|\boldsymbol{\theta}_i^t\|_1},$$

$$\hat{\boldsymbol{\theta}}_i^{t+1} \in \underset{\boldsymbol{\theta}_i \in \mathbb{R}_{\geq R^t}^{|\mathcal{A}_i|}}{\arg\min} \left\{ \langle -\boldsymbol{F}_i^t(\boldsymbol{\theta}^t), \boldsymbol{\theta}_i \rangle + \mathcal{B}_\psi(\boldsymbol{\theta}_i, \hat{\boldsymbol{\theta}}_i^t) \right\}, R^{t+1} = \begin{cases} R^t + 1 & \text{if } \|F^t(\boldsymbol{\theta}^t) - F^{t-1}(\boldsymbol{\theta}^{t-1})\|_2^2 \\ & \quad - \frac{\mathcal{B}_\psi(\hat{\boldsymbol{\theta}}^t, \boldsymbol{\theta}^{t-1}) + \mathcal{B}_\psi(\hat{\boldsymbol{\theta}}^t, \boldsymbol{\theta}^t)}{2} > 0, \\ R^t & \text{else} \end{cases} \quad (6)$$

where $R^1 = 1$, $\boldsymbol{\theta}_i^0 = \mathbf{1}/|A_i|$, $\hat{\boldsymbol{\theta}}_i^1 = \mathbf{1}/|A_i|$, $\boldsymbol{x}_i^0 = \mathbf{1}/|A_i|$, $\boldsymbol{F}_i^{t-1}(\boldsymbol{\theta}^{t-1}) = \langle \boldsymbol{\ell}_i^{t-1}, \boldsymbol{x}_i^{t-1} \rangle \mathbf{1} - \boldsymbol{\ell}_i^{t-1}$, $\boldsymbol{F}_i^t(\boldsymbol{\theta}^t) = \langle \boldsymbol{\ell}_i^t, \boldsymbol{x}_i^t \rangle \mathbf{1} - \boldsymbol{\ell}_i^t$, $\boldsymbol{\theta}^{t-1} = [\boldsymbol{\theta}_0^{t-1}; \boldsymbol{\theta}_1^{t-1}]$, $\boldsymbol{\theta}^t = [\boldsymbol{\theta}_0^t; \boldsymbol{\theta}_1^t]$, $\hat{\boldsymbol{\theta}}^t = [\hat{\boldsymbol{\theta}}_0^t; \hat{\boldsymbol{\theta}}_1^t]$, $\boldsymbol{F}^{t-1}(\boldsymbol{\theta}^{t-1}) = [\boldsymbol{F}_0^{t-1}(\boldsymbol{\theta}^{t-1}); \boldsymbol{F}_1^{t-1}(\boldsymbol{\theta}^{t-1})]$, $\boldsymbol{F}^t(\boldsymbol{\theta}^t) = [\boldsymbol{F}_0^t(\boldsymbol{\theta}^t); \boldsymbol{F}_1^t(\boldsymbol{\theta}^t)]$, and $\psi(\cdot)$ is the quadratic regularizer. The pseudocode for MI-SPRM$^+$ is in Algorithm 1.

The primary distinction between MI-SPRM$^+$ and SPRM$^+$ lies in the adaptive adjustment of the decision space (denoted as $\mathbb{R}_{\geq R^t}^{|\mathcal{A}_i|}$) that MI-SPRM$^+$ performs at each iteration $t$. This adaptation ensures that the lower bound for the 1-norm of accumulated regrets increases monotonically and exceeds $1/C_0$, since $R^t$ serves as such lower bound (from the definition of $\mathbb{R}_{\geq R^t}^{|\mathcal{A}_i|}$) and increases monotonically to a constant that exceeds $1/C_0$. According to these properties, MI-SPRM$^+$ obtains the $O(1/T)$ theoretical convergence rate, as shown in Theorem 4.1. See details in Section A.

**Theorem 4.1.** *[Proof is in Section A.] In a two-player zero-sum NFG, if all players employ MI-SPRM$^+$, then the weighted average strategy profile $\bar{\boldsymbol{x}}^T = \frac{(\sum_{t=1}^T R^t \boldsymbol{x}^t)}{(\sum_{t=1}^T R^t)}$ converges to an approximate NE with a rate of $O(1/T)$.*

**Discussion.** We now discuss whether our MI-SPRM$^+$ can be extended to multi-player general-sum NFGs or extensive-form games (EFGs). Firstly, regarding multi-player general-sum NFGs, it is crucial to clarify that, the complexity of computing a NE for multi-player general-sum NFGs belongs to the PPAD complexity class (Daskalakis et al., 2009). Therefore, no algorithm can achieve a polynomial-time convergence to an NE in such games. Our experiments further corroborate that none of the tested algorithms exhibited any convergence to NE when applied to multi-player general-sum NFGs. In fact, the original paper of SPRM$^+$ (Farina et al., 2023) only provides a social regret bound of $O(1)$ for multi-player general-sum NFGs, and does not offer any convergence rate to an NE in such games. As shown in Theorem A.1, we establish a similar social regret bound of $O(1)$ for multi-player general-sum NFGs: $(\sum_{t=1}^T R^t \langle \boldsymbol{\ell}^t, \boldsymbol{x}^t - \boldsymbol{x} \rangle)/(\sum_{t=1}^T R^t) \leq O(1)$. Secondly, for EFGs, the design of

MI-SPRM$^+$ can be directly extended to this domain. However, its $O(1/T)$ convergence rate does not hold. Specifically, RM variants are typically integrated with the Counterfactual Regret Minimization (CFR) framework to address EFGs. Unfortunately, to the best of my knowledge, only Clairvoyant CFR (Farina et al., 2023) achieves an $O(1/T)$ convergence rate when learning an NE of EFGs, albeit at the cost of an $O(\log T)$ per-iteration complexity (such complexity of our MI-SPRM$^+$ is $O(1)$). Experimental results show that the combination of MI-SPRM$^+$ and the CFR framework significantly outperforms other tested algorithms. In fact, such combination demonstrates an $O(1/T)$ or even faster empirical convergence rate.

## 5 EXPERIMENTS

**Configurations.** We now evaluate MI-SPRM$^+$ by comparing to RM$^+$ (Tammelin, 2014), PRM$^+$ (Farina et al., 2021), SPRM$^+$ (Farina et al., 2023), OGDA (Popov, 1980), OMWU (Rakhlin and Sridharan, 2013a), and DS-OptMD (Hsieh et al., 2021) (unless otherwise stated). Among them, MI-SPRM$^+$, RM$^+$, PRM$^+$, and DS-OptMD are parameter-free algorithms. Notably, although the update rules for RM$^+$ and PRM$^+$ include the step size $\eta$, the sequence of the strategy profiles $\boldsymbol{x}^1, \boldsymbol{x}^2, \cdots, \boldsymbol{x}^T$ that they generate remains unaffected by the value of $\eta$ (Farina et al., 2021), which is referred to as stepsize-invariance, also known as the strongly parameter-free property in Grand-Clément and Kroer (2021). This is why RM$^+$ and PRM$^+$ are referred to as parameter-free algorithms. MI-SPRM$^+$, SPRM$^+$, OGDA, and DS-OptMD achieve an $O(1/T)$ theoretical convergence rate while other tested algorithms only exhibit an $O(1/\sqrt{T})$ theoretical convergence rate. We use the duality gap as the metric to measure the distance to equilibrium. For non-parameter-free algorithms (SPRM$^+$ and OGDA), we choose step size $\eta$ from $[0.01, 0.1, 1]$. We do not use linear averaging and alternating updates. All experiments are conducted on a computer equipped with one Xeon(R) Gold 6444Y CPU and 256 GB of memory.

**Results on convergence rates in two-player zero-sum NFGs.** Now, we present the convergence results in two-player zero-sum NFGs. We conduct experiments on the $3 \times 3$ two-player zero-sum NFGs considered in the original paper of SPRM$^+$ (Farina et al., 2023), whose payoff matrix is $[[3, 0, -3], [0, 3, -4], [0, 0, 1]]$, and randomly generated two-player zero-sum NFGs of varying sizes: $[10, 30, 50, 100]$. For each size, we generate 60 independent instances to ensure the robustness of our results. Specifically, for each set of 20 instances, the payoff matrices are drawn from distinct Gaussian distributions. The first group uses a Gaussian distribution with a mean of 0 and a standard deviation of 1, the second group employs a Gaussian distribution with a mean of 0 and a standard deviation of 10, and the final group uses a Gaussian distribution with a mean of 0 and a standard deviation of 100. We present the average duality gaps across the 20 instances for each group and report the corresponding confidence intervals.

The convergence rates on the $3 \times 3$ two-player zero-sum NFGs considered in the original paper of SPRM$^+$ (Farina et al., 2023) are demonstrated in Fig. 1. The experimental results demonstrate that SPRM$^+$ is highly sensitive to the step size parameter, with performance variations of up to tenfold depending on the chosen value. In contrast, MI-SPRM$^+$ eliminates the need for parameter tuning and achieves a faster convergence rate compared to SPRM$^+$. The convergence results on randomly generated two-player zero-sum NFGs are shown in Figs. 2, 4, and 5 (due to page limitations, Figs. 4 and 5 are included in Section C). MI-SPRM$^+$ achieves an $O(1/T)$ empirical convergence rate across all tested games. Notably, MI-SPRM$^+$ outperforms all other algorithms. Additionally, our findings indicate that the traditional regret minimization algorithms, such as OGDA and OMWU, are more sensitive compared to RM$^+$ variants. Specifically, in games where payoff matrices

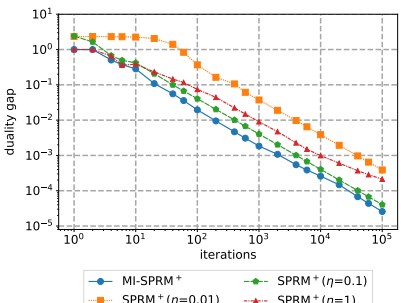

Figure 1: Convergence rates of MI-SPRM$^+$ and SPRM$^+$ with different step sizes $\eta$ in the $3 \times 3$ two-player zero-sum NFGs considered in the original paper of SPRM$^+$. The duality gaps at iteration 1e5 for MI-SPRM$^+$, SPRM$^+$ ($\eta = 0.01$), SPRM$^+$ ($\eta = 0.1$), and SPRM$^+$ ($\eta = 1$) are 2.6e-5, 3.9e-4, 4.0e-5, and 2.1e-4, respectively.

are sampled from a Gaussian distribution with mean 0 and standard deviation 1, as well as a Gaussian distribution with mean 0 and standard deviation 10, OGDA and OMWU only converge when $\eta$ is

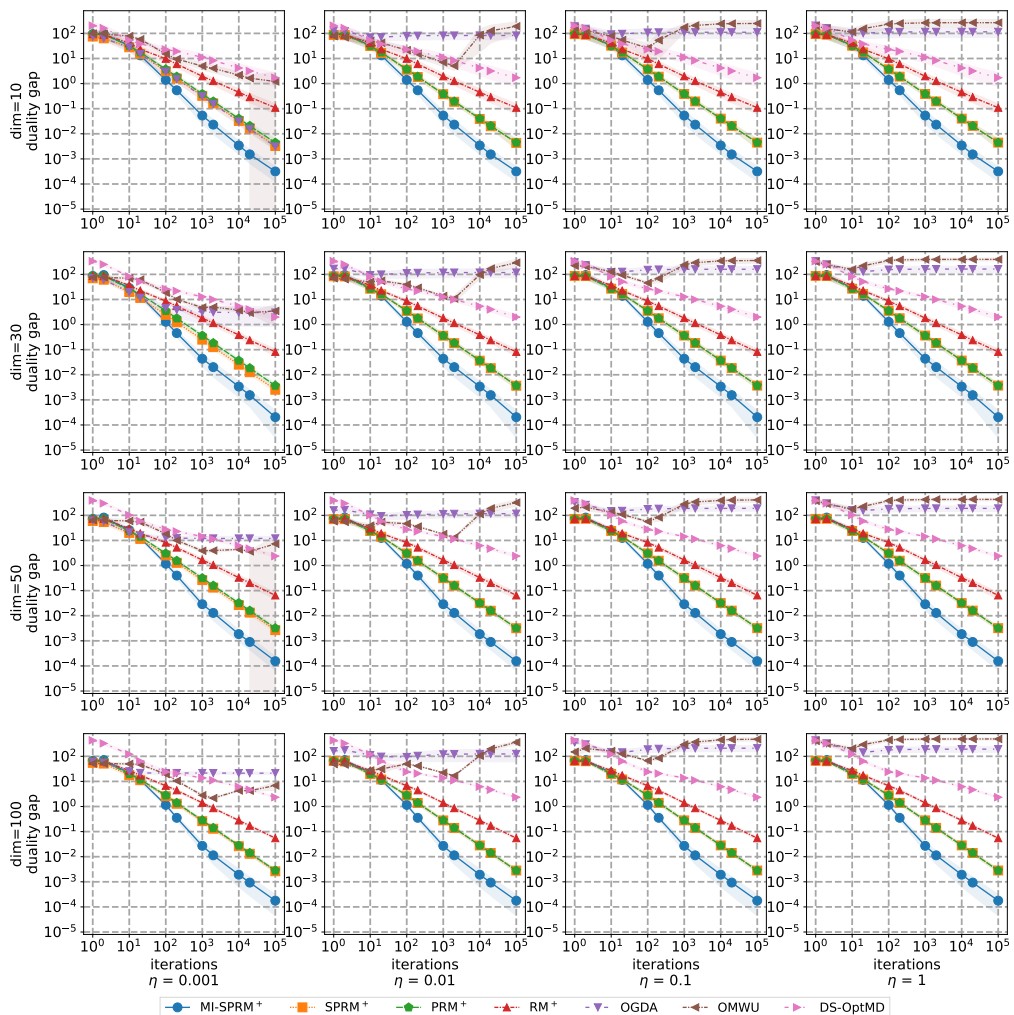

Figure 2: Convergence rates of different algorithms in randomly generated two-player zero-sum NFGs, where payoff matrices are sampled from a Gaussian distribution with mean 0 and standard deviation 100. Note that the value of $\eta$ only involves the performance of SPRM$^+$ and OGDA as other algorithms are parameter-free algorithms.

sufficiently small. In games where payoff matrices are sampled from a Gaussian distribution with mean 0 and standard deviation 100, OGDA and OMWU only converges when $\eta$ is 0.001 and the dimension of the game is less than 10.

It is important to note that the results in Fig. 5 may exhibit slight scale distortion. In fact, in terms of the average reduction in duality gap, MI-SPRM$^+$ demonstrates approximately a 37% improvement compared to SPRM$^+$ (the best-performing algorithm excluding our MI-SPRM$^+$). This reduction is comparable to the 42% improvement in Xu et al. (2024a) (which also investigate RM variants). Furthermore, in the games considered in Figs. 2 and 4, MI-SPRM$^+$ achieves a remarkable reduction of 92% and 74%, respectively, relative to SPRM$^+$. These reductions significantly surpass the reductions in Xu et al. (2024a).

Moreover, although DS-OptMD theoretically guarantees an $O(1/T)$ convergence rate, it fails to empirically demonstrate this rate. We argue that this is because it achieves an $O(1/T)$ theoretical convergence rate through adaptive step size reduction. However, a substantial number of iterations is necessary to sufficiently reduce the step size and fully realize the $O(1/T)$ convergence rate. For a more detailed discussion on the empirical convergence rate of DS-OptMD, refer to the paragraph titled "Results on the dynamics of the values of $R^t$" in Section C.

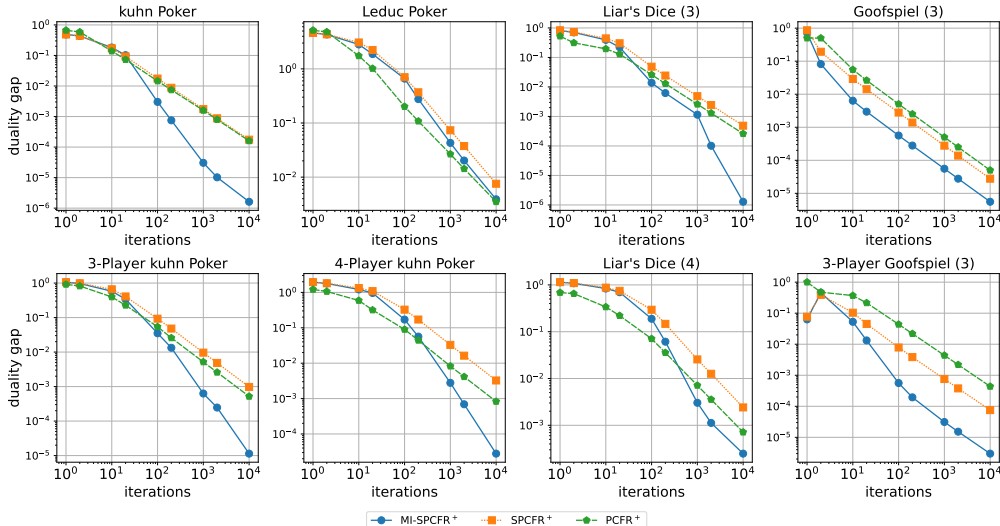

Figure 3: Convergence rates of different algorithms in standard EFG benchmarks.

**Results on convergence rates in multi-player general-sum NFGs.** We also evaluate the performance of MI-SPRM$^+$, RM$^+$, PRM$^+$, SPRM$^+$, OGDA, OMWU, and DS-OptMD in multi-player general-sum NFGs. The results are show in Section C (Fig. 6). Consistent with theory, no algorithm can learn an NE in all tested multi-player general-sum NFGs. See more details in Section C.

**Results on convergence rates in EFGs.** Now, we evaluate the performance of MI-SPRM$^+$ in EFGs. We test on eight instances of four standard EFG benchmarks: Kuhn Poker, Leduc Poker, Liar's Dice, and Goofspiel. These EFGs are implemented using OpenSpiel (Lanctot et al., 2019). For Kuhn Poker, we examine the two-player, three-player, and four-player versions, denoted as "Kuhn Poker", "3-Player Kuhn Poker", and "4-Player Kuhn Poker", respectively. In the case of Leduc Poker, only its two-player version is tested due to the size constraints of the three-player variant. For Liar's Dice, OpenSpiel's limitations prevent testing of versions with three or more players; therefore, we analyze the versions with 3 and 4 sides, denoted as "Liar's Dice (3)" and "Liar's Dice (4)", respectively. Lastly, for Goofspiel, we set the number of cards to 3 and test both the two-player and three-player versions, referred to as "Goofspiel (3)" and "3-Player Goofspiel (3)". As RM variants are typically integrated with the CFR framework to address EFGs, we integrate MI-SPRM$^+$ with the CFR framework and get MI-SPCFR$^+$. We compare MI-SPCFR$^+$ against the combination of SPRM$^+$ with the CFR framework, referred to as SPCFR$^+$, and the combination of PRM$^+$ with the CFR framework, known as PCFR$^+$ (Farina et al., 2021). We do not to compare with other algorithms tested in our NFGs experiments as they consistently underperform MI-SPRM$^+$, SPRM$^+$, and PRM$^+$. The results are shown in Fig. 3. We observe that MI-SPCFR$^+$ significantly surpasses SPCFR$^+$ in all eight tested games. In addition, although PCFR$^+$ outperforms MI-SPCFR$^+$ in Leduc Poker, MI-SPCFR$^+$ significantly outperforms PCFR$^+$ in the remaining seven tested games. Moreover, the experimental results demonstrate that our algorithm achieves an $O(1/T)$ or even faster empirical convergence rate even on multi-player EFGs. Interestingly, while an $O(1/T)$ or even faster empirical convergence rate is observed in multi-player EFGs, no such empirical convergence is noted in multi-player NFGs. We hypothesize that this arises due to the unique characteristics of the CFR framework and the tested EFGs–Kuhn Poker and Goofspiel. However, this remains an open question, as no existing work has provided a theoretical explanation to date.

**Results on runtimes.** We compare the runtimes of MI-SPRM$^+$, SPRM$^+$, and DS-OptMD, as shown in Table 1, in randomly generated two-player zero-sum NFGs. For each game dimension, we average the runtimes over 60 instances. Although all three algorithms have the same theoretical per-iteration complexity, $O\left(\sum_{i\in\mathcal{N}} |\mathcal{A}_i| \log |\mathcal{A}_i|\right)$, both MI-SPRM$^+$ and DS-OptMD require parameter learning to achieve their parameter-free properties, resulting in longer runtimes compared to SPRM$^+$. Specifically, the runtime of MI-SPRM$^+$ is approximately 1.5 times that of SPRM$^+$, while DS-OptMD's runtime is about 2.5 times longer. We hypothesize that the significantly higher runtime of

Table 1: Comparison of the runtime (in minutes) between MI-SPRM$^+$, SPRM$^+$, and DS-OptMD for randomly generated two-player zero-sum NFGs. It is important to highlight that theoretical per-iteration complexity for MI-SPRM$^+$, SPRM$^+$, and DS-OptMD remains $O\left(\sum_{i\in\mathcal{N}}|\mathcal{A}_i|\log|\mathcal{A}_i|\right)$.

|  | MI-SPRM$^+$ | SPRM$^+$ | DS-OptMD |
|---|---|---|---|
| dim=10 | 0.1105±0.0007 | 0.0751±0.0016 | 0.1746±0.0007 |
| dim=30 | 0.1367±0.0009 | 0.0858±0.0007 | 0.2001±0.0011 |
| dim=50 | 0.1642±0.0016 | 0.1023±0.0016 | 0.2462±0.0005 |
| dim=100 | 0.2398±0.0014 | 0.1562±0.0013 | 0.3717±0.0018 |

DS-OptMD stems from its requirement for individual parameter learning for each player, a step that is circumvented in both MI-SPRM$^+$ and SPRM$^+$. A key direction for future research is to reduce the time required for parameter learning while preserving the parameter-free property.

## 6 CONCLUSIONS

In this paper, we investigate parameter-free RM variants. To the best of our knowledge, we propose the first parameter-free RM variant that achieves an $O(1/T)$ theoretical convergence rate, named MI-SPRM$^+$. To achieve the parameter-free property and $O(1/T)$ theoretical convergence rate simultaneously, MI-SPRM$^+$ ensures that the lower bound for the 1-norm of accumulated regrets monotonically increases by adjusting the decision space at each iteration. The empirical results indicate that MI-SPRM$^+$ attains an empirical convergence rate of $O(1/T)$ in all tested games, and MI-SPRM$^+$ outperforms all other tested algorithms, including existing RM variants, and traditional regret minimization algorithms. By combining MI-SPRM$^+$ with the CFR framework, we get MI-SPCFR$^+$, which outperforms other classical CFR algorithms like PCFR$^+$, as shown in our experimental results.

## ACKNOWLEDGEMENTS

This work is supported by Shanghai Artificial Intelligence Laboratory, the National Natural Science Foundation of China under Grant 62192783, the Jiangsu Science and Technology Major Project BG2024031, the Fundamental Research Funds for the Central Universities (14380128), the Collaborative Innovation Center of Novel Software Technology and Industrialization, and the InnoHK funding.

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

# A PROOF OF THEOREM 4.1

To prove Theorem 4.1, we introduce Theorem A.1.

**Lemma A.1.** *[Proof is in Section A.1.] In a multi-player general-sum NFG, if all players employ MI-SPRM$^+$, then the weighted social regret bound is bounded by*

$$\frac{\sum_{t=1}^{T} R^t \langle \boldsymbol{\ell}^t, \boldsymbol{x}^t - \boldsymbol{x} \rangle}{\sum_{t=1}^{T} R^t} \leq O(1).$$

*Proof.* According to folk theorem, Theorem A.1 implies that the weighted average strategy profile $\bar{\boldsymbol{x}}^T = (\sum_{t=1}^{T} R^t \boldsymbol{x}^t)/(\sum_{t=1}^{T} R^t)$ converges to an approximate NE with a rate of $O(1/T)$ since $\sum_{t=1}^{T} R^t \langle \boldsymbol{\ell}^t, \boldsymbol{x}^t - \boldsymbol{x} \rangle$ can be interpreted as the social regret over a newly sequence of strategy profiles, $\{\boldsymbol{x}^1, \ldots, \underbrace{\boldsymbol{x}^t, \ldots, \boldsymbol{x}^t}_{R^t}, \ldots\}$. It completes the proof. $\square$

## A.1 PROOF OF LEMMA A.1

*Proof.* To prove Theorem A.1, we first prove that for any initial $R^1$ (defined in Eq. (6) with $t = 1$), $R^t$ increases monotonically to a constant as $t \to \infty$. By using this property, we show that $\sum_{t=1}^{T} R^t \langle \boldsymbol{\ell}^t, \boldsymbol{x}^t - \boldsymbol{x} \rangle \leq O(1)$ for any $T \geq 1$.

Before starting our proofs, we introduce Lemmas A.2 and A.3, which are very important for our proofs.

**Lemma A.2** (Adapted from Lemma 10 of Wei et al. (2021)). *Let $\mathcal{A}$ as a convex set and $\boldsymbol{a}' \in \arg\min_{\boldsymbol{a}' \in \mathcal{A}}\{\langle \boldsymbol{a}', \boldsymbol{g} \rangle + \mathcal{B}_\psi(\boldsymbol{a}', \boldsymbol{a})\}$. Then for any $\boldsymbol{a}^* \in \mathcal{A}$, we have*

$$\langle \boldsymbol{a}' - \boldsymbol{a}^*, \boldsymbol{g} \rangle \leq \mathcal{B}_\psi(\boldsymbol{a}^*, \boldsymbol{a}) - \mathcal{B}_\psi(\boldsymbol{a}^*, \boldsymbol{a}') - \mathcal{B}_\psi(\boldsymbol{a}', \boldsymbol{a}),$$

*where $\psi(\cdot)$ is the quadratic regularizer defined in Eq. (3).*

**Lemma A.3** (Proof is in Section A.2). *Suppose $\|\boldsymbol{\theta}_i^t\|_1$ and $\|\boldsymbol{\theta}_i^{t-1}\|_1$ are greater than a constant $C_1$ for all player $i$, we have*

$$\|\boldsymbol{F}^t(\boldsymbol{\theta}^t) - \boldsymbol{F}^{t-1}(\boldsymbol{\theta}^{t-1})\|_2^2 \leq \frac{2C_2}{C_1^2}\Big(\mathcal{B}_\psi(\hat{\boldsymbol{\theta}}^t, \boldsymbol{\theta}^{t-1}) + \mathcal{B}_\psi(\hat{\boldsymbol{\theta}}^t, \boldsymbol{\theta}^t)\Big),$$

*where $\psi(\cdot)$ is the quadratic regularizer defined in Eq. (3), and $C_2 = 2D(2L^2 + 4DL^2 + 4DP^2)$.*

Now, we first prove that for initial $R^1$ (not only 1), $R^t$ increases monotonically to a constant as $t \to \infty$. By using Lemma A.3 with $C_1 = R^{t-1}$, we have (from the update rule of MI-SPRM$^+$ as shown in Eq. (6), $R^t \geq R^{t-1}$)

$$\|\boldsymbol{F}^t(\boldsymbol{\theta}^t) - \boldsymbol{F}^{t-1}(\boldsymbol{\theta}^{t-1})\|_2^2 \leq \frac{2C_2}{(R^{t-1})^2}\Big(\mathcal{B}_\psi(\hat{\boldsymbol{\theta}}^t, \boldsymbol{\theta}^{t-1}) + \mathcal{B}_\psi(\hat{\boldsymbol{\theta}}^t, \boldsymbol{\theta}^t)\Big).$$

If $R^{t-1} \geq 2\sqrt{C_2}$, we can obtain

$$\|\boldsymbol{F}^t(\boldsymbol{\theta}^t) - \boldsymbol{F}^{t-1}(\boldsymbol{\theta}^{t-1})\|_2^2 \leq \frac{2C_2}{(R^{t-1})^2}\Big(\mathcal{B}_\psi(\hat{\boldsymbol{\theta}}^t, \boldsymbol{\theta}^{t-1}) + \mathcal{B}_\psi(\hat{\boldsymbol{\theta}}^t, \boldsymbol{\theta}^t)\Big) \leq \frac{2C_2}{4C_2}\Big(\mathcal{B}_\psi(\hat{\boldsymbol{\theta}}^t, \boldsymbol{\theta}^{t-1}) + \mathcal{B}_\psi(\hat{\boldsymbol{\theta}}^t, \boldsymbol{\theta}^t)\Big)$$
$$\leq \frac{1}{2}\Big(\mathcal{B}_\psi(\hat{\boldsymbol{\theta}}^t, \boldsymbol{\theta}^{t-1}) + \mathcal{B}_\psi(\hat{\boldsymbol{\theta}}^t, \boldsymbol{\theta}^t)\Big), \tag{7}$$

implying that $\|\boldsymbol{F}^t(\boldsymbol{\theta}^t) - \boldsymbol{F}^t(\boldsymbol{\theta}^{t-1})\|_2^2 - \Big(\mathcal{B}_\psi(\hat{\boldsymbol{\theta}}^t, \boldsymbol{\theta}^{t-1}) + \mathcal{B}_\psi(\hat{\boldsymbol{\theta}}^t, \boldsymbol{\theta}^t)\Big)/2 \leq 0$ always holds. Therefore, from Eq. (6) and (7), we have that $R^t$ increases monotonically and once $R^{t-1} \geq 2\sqrt{C_2}$, for any $t' \geq t-1$, $R^{t'} = R^{t-1}$. Therefore, for any $R^1$, $R^t$ increases monotonically to a constant $C_3$ as $t \to \infty$. Note that $C_3 \geq \max(R_1, 2\sqrt{C_2}) = \max(R_1, \sqrt{8D(2L^2 + 4DL^2 + 4DP^2)}) = \max(R_1, 1/C_0)$ (since $C_0 = 1/\sqrt{8D(2L^2 + 4DL^2 + 4DP^2)}$) as $t \to \infty$.

Considering the third line of Eq. (6), and using Lemma A.2 with $\boldsymbol{a} = \hat{\boldsymbol{\theta}}^t = [\hat{\boldsymbol{\theta}}_0^t; \hat{\boldsymbol{\theta}}_1^t]$, $\boldsymbol{a}' = \hat{\boldsymbol{\theta}}^{t+1} = [\hat{\boldsymbol{\theta}}_0^{t+1}; \hat{\boldsymbol{\theta}}_1^{t+1}]$, $\boldsymbol{a}^* = \boldsymbol{\theta} = [\boldsymbol{\theta}_0; \boldsymbol{\theta}_1]$ and $\boldsymbol{g} = -\boldsymbol{F}^t(\boldsymbol{\theta}^t) = [-\boldsymbol{F}_0^t(\boldsymbol{\theta}^t); -\boldsymbol{F}_1^t(\boldsymbol{\theta}^t)]$ (in this case, $\mathcal{A}$ is $\times_{i \in \mathcal{N}} \mathbb{R}_{\geq R^t}^{|\mathcal{A}_i|}$, which means $\boldsymbol{\theta} \in \times_{i \in \mathcal{N}} \mathbb{R}_{\geq R^t}^{|\mathcal{A}_i|}$), we have

$$\langle -\boldsymbol{F}^t(\boldsymbol{\theta}^t), \hat{\boldsymbol{\theta}}^{t+1} - \boldsymbol{\theta} \rangle \leq \mathcal{B}_\psi(\boldsymbol{\theta}, \hat{\boldsymbol{\theta}}^t) - \mathcal{B}_\psi(\boldsymbol{\theta}, \hat{\boldsymbol{\theta}}^{t+1}) - \mathcal{B}_\psi(\hat{\boldsymbol{\theta}}^{t+1}, \hat{\boldsymbol{\theta}}^t). \tag{8}$$

Similarly, considering the first line of Eq. (6), and using Lemma A.2 with $\boldsymbol{a} = \hat{\boldsymbol{\theta}}^t = [\hat{\boldsymbol{\theta}}_0^t; \hat{\boldsymbol{\theta}}_1^t]$, $\boldsymbol{a}' = \boldsymbol{\theta}^t = [\boldsymbol{\theta}_0^t; \boldsymbol{\theta}_1^t]$, $\boldsymbol{a}^* = \hat{\boldsymbol{\theta}}^{t+1} = [\hat{\boldsymbol{\theta}}_0^{t+1}; \hat{\boldsymbol{\theta}}_1^{t+1}]$ and $\boldsymbol{g} = -\boldsymbol{F}^{t-1}(\boldsymbol{\theta}^{t-1}) = [-\boldsymbol{F}_0^{t-1}(\boldsymbol{\theta}^{t-1}); -\boldsymbol{F}_1^{t-1}(\boldsymbol{\theta}^{t-1})]$, we get

$$\langle -\boldsymbol{F}^{t-1}(\boldsymbol{\theta}^{t-1}), \boldsymbol{\theta}^t - \hat{\boldsymbol{\theta}}^{t+1}\rangle \leq \mathcal{B}_\psi(\hat{\boldsymbol{\theta}}^{t+1}, \hat{\boldsymbol{\theta}}^t) - \mathcal{B}_\psi(\hat{\boldsymbol{\theta}}^{t+1}, \boldsymbol{\theta}^t) - \mathcal{B}_\psi(\boldsymbol{\theta}^t, \hat{\boldsymbol{\theta}}^t). \tag{9}$$

Summing up Eq. (8) and (9), and adding $\langle \boldsymbol{F}^t(\boldsymbol{\theta}^t) - \boldsymbol{F}^{t-1}(\boldsymbol{\theta}^{t-1}), \hat{\boldsymbol{\theta}}^{t+1} - \boldsymbol{\theta}^t\rangle$ to both sides, we get

$$\langle -\boldsymbol{F}^t(\boldsymbol{\theta}^t), \boldsymbol{\theta}^t - \boldsymbol{\theta}\rangle \leq \mathcal{B}_\psi(\boldsymbol{\theta}, \hat{\boldsymbol{\theta}}^t) - \mathcal{B}_\psi(\boldsymbol{\theta}, \hat{\boldsymbol{\theta}}^{t+1}) - \mathcal{B}_\psi(\hat{\boldsymbol{\theta}}^{t+1}, \boldsymbol{\theta}^t) - \mathcal{B}_\psi(\boldsymbol{\theta}^t, \hat{\boldsymbol{\theta}}^t)$$
$$+ \langle \boldsymbol{F}^t(\boldsymbol{\theta}^t) - \boldsymbol{F}^{t-1}(\boldsymbol{\theta}^{t-1}), \hat{\boldsymbol{\theta}}^{t+1} - \boldsymbol{\theta}^t\rangle. \tag{10}$$

For the term $\langle \boldsymbol{F}^t(\boldsymbol{\theta}^t), \boldsymbol{\theta}^t\rangle$, we get

$$\langle \boldsymbol{F}^t(\boldsymbol{\theta}^t), \boldsymbol{\theta}^t\rangle = \sum_{i\in\mathcal{N}} \langle \boldsymbol{F}_i^t(\boldsymbol{\theta}^t), \boldsymbol{\theta}_i^t\rangle = \sum_{i\in\mathcal{N}} \langle \langle \boldsymbol{\ell}_i^t, \boldsymbol{x}_i^t\rangle \mathbf{1} - \boldsymbol{\ell}_i^t, \boldsymbol{\theta}_i^t\rangle = \sum_{i\in\mathcal{N}} \left( \langle \boldsymbol{\ell}_i^t, \boldsymbol{x}_i^t\rangle \langle \mathbf{1}, \boldsymbol{\theta}_i^t\rangle - \langle \boldsymbol{\ell}_i^t, \boldsymbol{\theta}_i^t\rangle\right)$$
$$= \sum_{i\in\mathcal{N}} \left( \langle \boldsymbol{\ell}_i^t, \frac{\boldsymbol{\theta}_i^t}{\|\boldsymbol{\theta}_i^t\|_1}\rangle \|\boldsymbol{\theta}_i^t\|_1 - \langle \boldsymbol{\ell}_i^t, \boldsymbol{\theta}_i^t\rangle\right) = 0, \tag{11}$$

where the last equality is from $\boldsymbol{x}_i^t = \boldsymbol{\theta}_i^t/\|\boldsymbol{\theta}_i^t\|_1$ (as stated in Eq. (6)). Arranging the terms of Eq. (10) and using the fact in Eq. (11), we have

$$\mathcal{B}_\psi(\boldsymbol{\theta}, \hat{\boldsymbol{\theta}}^{t+1}) - \mathcal{B}_\psi(\boldsymbol{\theta}, \hat{\boldsymbol{\theta}}^t)$$
$$\leq -\langle \boldsymbol{F}^t(\boldsymbol{\theta}^t), \boldsymbol{\theta}\rangle + \langle \boldsymbol{F}^t(\boldsymbol{\theta}^t) - \boldsymbol{F}^{t-1}(\boldsymbol{\theta}^{t-1}), \hat{\boldsymbol{\theta}}^{t+1} - \boldsymbol{\theta}^t\rangle - \mathcal{B}_\psi(\hat{\boldsymbol{\theta}}^{t+1}, \boldsymbol{\theta}^t) - \mathcal{B}_\psi(\boldsymbol{\theta}^t, \hat{\boldsymbol{\theta}}^t). \tag{12}$$

Substituting $\boldsymbol{\theta} = R^t\boldsymbol{x}$, $\boldsymbol{x} \in \mathcal{X}$ (note that $\boldsymbol{\theta} \in \times_{i\in\mathcal{N}}\mathbb{R}_{\geq R^t}^{|\mathcal{A}_i|}$) into Eq. (12), we have

$$\mathcal{B}_\psi(R^t\boldsymbol{x}, \hat{\boldsymbol{\theta}}^{t+1}) - \mathcal{B}_\psi(R^t\boldsymbol{x}, \hat{\boldsymbol{\theta}}^t)$$
$$\leq -\langle \boldsymbol{F}^t(\boldsymbol{\theta}^t), R^t\boldsymbol{x}\rangle + \langle \boldsymbol{F}^t(\boldsymbol{\theta}^t) - \boldsymbol{F}^{t-1}(\boldsymbol{\theta}^{t-1}), \hat{\boldsymbol{\theta}}^{t+1} - \boldsymbol{\theta}^t\rangle - \mathcal{B}_\psi(\hat{\boldsymbol{\theta}}^{t+1}, \boldsymbol{\theta}^t) - \mathcal{B}_\psi(\boldsymbol{\theta}^t, \hat{\boldsymbol{\theta}}^t) \tag{13}$$
$$\leq -R^t\langle \boldsymbol{\ell}^t, \boldsymbol{x}^t - \boldsymbol{x}\rangle + \langle \boldsymbol{F}^t(\boldsymbol{\theta}^t) - \boldsymbol{F}^{t-1}(\boldsymbol{\theta}^{t-1}), \hat{\boldsymbol{\theta}}^{t+1} - \boldsymbol{\theta}^t\rangle - \mathcal{B}_\psi(\hat{\boldsymbol{\theta}}^{t+1}, \boldsymbol{\theta}^t) - \mathcal{B}_\psi(\boldsymbol{\theta}^t, \hat{\boldsymbol{\theta}}^t),$$

where the last inequality comes from $\langle \boldsymbol{F}^t(\boldsymbol{\theta}^t), R^t\boldsymbol{x}\rangle = \sum_{i\in\mathcal{N}} \langle \langle \boldsymbol{\ell}_i^t, \boldsymbol{x}_i^t\rangle \mathbf{1} - \boldsymbol{\ell}_i^t, R^t\boldsymbol{x}_i\rangle = R^t\langle \boldsymbol{\ell}^t, \boldsymbol{x}^t - \boldsymbol{x}\rangle$ ($\boldsymbol{x}_i \in \mathcal{X}_i$ implies $\langle \mathbf{1}, \boldsymbol{x}_i\rangle = 1$, as stated around Eq.(1)). Arranging the terms in Eq. (13), we get

$$R^t\langle \boldsymbol{\ell}^t, \boldsymbol{x}^t - \boldsymbol{x}\rangle + \mathcal{B}_\psi(R^t\boldsymbol{x}, \hat{\boldsymbol{\theta}}^{t+1}) - \mathcal{B}_\psi(R^t\boldsymbol{x}, \hat{\boldsymbol{\theta}}^t)$$
$$\leq \langle \boldsymbol{F}^t(\boldsymbol{\theta}^t) - \boldsymbol{F}^{t-1}(\boldsymbol{\theta}^{t-1}), \hat{\boldsymbol{\theta}}^{t+1} - \boldsymbol{\theta}^t\rangle - \mathcal{B}_\psi(\hat{\boldsymbol{\theta}}^{t+1}, \boldsymbol{\theta}^t) - \mathcal{B}_\psi(\boldsymbol{\theta}^t, \hat{\boldsymbol{\theta}}^t)$$
$$\leq \|\boldsymbol{F}^t(\boldsymbol{\theta}^t) - \boldsymbol{F}^{t-1}(\boldsymbol{\theta}^{t-1})\|_2 \|\hat{\boldsymbol{\theta}}^{t+1} - \boldsymbol{\theta}^t\|_2 - \mathcal{B}_\psi(\hat{\boldsymbol{\theta}}^{t+1}, \boldsymbol{\theta}^t) - \mathcal{B}_\psi(\boldsymbol{\theta}^t, \hat{\boldsymbol{\theta}}^t)$$
$$\leq \frac{2\|\boldsymbol{F}^t(\boldsymbol{\theta}^t) - \boldsymbol{F}^{t-1}(\boldsymbol{\theta}^{t-1})\|_2^2}{2} + \frac{\|\hat{\boldsymbol{\theta}}^{t+1} - \boldsymbol{\theta}^t\|_2^2}{2\times 2} - \mathcal{B}_\psi(\hat{\boldsymbol{\theta}}^{t+1}, \boldsymbol{\theta}^t) - \mathcal{B}_\psi(\boldsymbol{\theta}^t, \hat{\boldsymbol{\theta}}^t) \tag{14}$$
$$= \|\boldsymbol{F}^t(\boldsymbol{\theta}^t) - \boldsymbol{F}^{t-1}(\boldsymbol{\theta}^{t-1})\|_2^2 - \frac{\mathcal{B}_\psi(\hat{\boldsymbol{\theta}}^{t+1}, \boldsymbol{\theta}^t)}{2} - \mathcal{B}_\psi(\boldsymbol{\theta}^t, \hat{\boldsymbol{\theta}}^t)$$
$$\leq \|\boldsymbol{F}^t(\boldsymbol{\theta}^t) - \boldsymbol{F}^{t-1}(\boldsymbol{\theta}^{t-1})\|_2^2 - \frac{\mathcal{B}_\psi(\hat{\boldsymbol{\theta}}^{t+1}, \boldsymbol{\theta}^t)}{2} - \frac{\mathcal{B}_\psi(\boldsymbol{\theta}^t, \hat{\boldsymbol{\theta}}^t)}{2},$$

where the third inequality is from that $ab \leq \rho b^2/2 + c^2/(2\rho), \forall b, c, \rho > 0$ (here, $b = \|\boldsymbol{F}^t(\boldsymbol{\theta}^t) - \boldsymbol{F}^{t-1}(\boldsymbol{\theta}^{t-1})\|_2$, $c = \|\hat{\boldsymbol{\theta}}^{t+1} - \boldsymbol{\theta}^t\|_2$, and $\rho = 2$), and the last equality is from $\mathcal{B}_\psi(\boldsymbol{a}, \boldsymbol{b}) = \|\boldsymbol{a} - \boldsymbol{b}\|_2^2/2$ ($\psi(\cdot)$ is the quadratic regularizer as stated around Eq. (6), as well as $\mathcal{B}_\psi(\boldsymbol{a}, \boldsymbol{b}) = \|\boldsymbol{a} - \boldsymbol{b}\|_2^2/2$ if $\psi(\cdot)$ is the quadratic regularizer as stated around Eq. (3)). Summing up Eq. (14) from $t = 1$ to $T$, we have

$$\sum_{t=1}^T R^t\langle \boldsymbol{\ell}^t, \boldsymbol{x}^t - \boldsymbol{x}\rangle + \mathcal{B}_\psi(R^T\boldsymbol{x}, \hat{\boldsymbol{\theta}}^{T+1}) - \mathcal{B}_\psi(R^1\boldsymbol{x}, \hat{\boldsymbol{\theta}}^1) + \sum_{t=2}^T \left( -\mathcal{B}_\psi(R^t\boldsymbol{x}, \hat{\boldsymbol{\theta}}^t) + \mathcal{B}_\psi(R^{t-1}\boldsymbol{x}, \hat{\boldsymbol{\theta}}^t)\right)$$
$$\leq \sum_{t=1}^T \|\boldsymbol{F}^t(\boldsymbol{\theta}^t) - \boldsymbol{F}^{t-1}(\boldsymbol{\theta}^{t-1})\|_2^2 - \frac{\mathcal{B}_\psi(\hat{\boldsymbol{\theta}}^{T+1}, \boldsymbol{\theta}^T)}{2} - \sum_{t=1}^T \left( \frac{\mathcal{B}_\psi(\boldsymbol{\theta}^t, \hat{\boldsymbol{\theta}}^t)}{2} + \frac{\mathcal{B}_\psi(\hat{\boldsymbol{\theta}}^t, \boldsymbol{\theta}^{t-1})}{2}\right) + \frac{\mathcal{B}_\psi(\hat{\boldsymbol{\theta}}^1, \boldsymbol{\theta}^0)}{2}$$
$$\leq \sum_{t=1}^T \|\boldsymbol{F}^t(\boldsymbol{\theta}^t) - \boldsymbol{F}^{t-1}(\boldsymbol{\theta}^{t-1})\|_2^2 - \sum_{t=1}^T \left( \frac{\mathcal{B}_\psi(\boldsymbol{\theta}^t, \hat{\boldsymbol{\theta}}^t)}{2} + \frac{\mathcal{B}_\psi(\hat{\boldsymbol{\theta}}^t, \boldsymbol{\theta}^{t-1})}{2}\right) + \frac{\mathcal{B}_\psi(\hat{\boldsymbol{\theta}}^1, \boldsymbol{\theta}^0)}{2}, \tag{15}$$

where the first line is from that $\sum_{t=1}^T (\mathcal{B}_\psi(R^t\boldsymbol{x}, \hat{\boldsymbol{\theta}}^{t+1}) - \mathcal{B}_\psi(R^t\boldsymbol{x}, \hat{\boldsymbol{\theta}}^t)) = \mathcal{B}_\psi(R^T\boldsymbol{x}, \hat{\boldsymbol{\theta}}^{T+1}) - \mathcal{B}_\psi(R^1\boldsymbol{x}, \hat{\boldsymbol{\theta}}^1) + \sum_{t=2}^T (-\mathcal{B}_\psi(R^t\boldsymbol{x}, \hat{\boldsymbol{\theta}}^t) + \mathcal{B}_\psi(R^{t-1}\boldsymbol{x}, \hat{\boldsymbol{\theta}}^t))$, and the first inequality is from

that $\sum_{t=1}^{T}(-\mathcal{B}_\psi(\hat{\boldsymbol{\theta}}^{t+1},\boldsymbol{\theta}^t)/2 - \mathcal{B}_\psi(\boldsymbol{\theta}^t,\hat{\boldsymbol{\theta}}^t)/2) = -\mathcal{B}_\psi(\hat{\boldsymbol{\theta}}^{T+1},\boldsymbol{\theta}^T)/2 - \sum_{t=1}^{T}(\mathcal{B}_\psi(\boldsymbol{\theta}^t,\hat{\boldsymbol{\theta}}^t)/2 + \mathcal{B}_\psi(\hat{\boldsymbol{\theta}}^t,\boldsymbol{\theta}^{t-1})/2) + \mathcal{B}_\psi(\hat{\boldsymbol{\theta}}^1,\boldsymbol{\theta}^0)/2$. In addition, for the term $\sum_{t=2}^{T}(-\mathcal{B}_\psi(R^t\boldsymbol{x},\hat{\boldsymbol{\theta}}^t) + \mathcal{B}_\psi(R^{t-1}\boldsymbol{x},\hat{\boldsymbol{\theta}}^t))$, as $\psi(\cdot)$ is the quadratic regularizer (as stated around Eq. (6)), we get

$$\sum_{t=2}^{T}\Big(-\mathcal{B}_\psi(R^t\boldsymbol{x},\hat{\boldsymbol{\theta}}^t) + \mathcal{B}_\psi(R^{t-1}\boldsymbol{x},\hat{\boldsymbol{\theta}}^t)\Big)$$
$$=\sum_{t=2}^{T}\big((R^{t-1})^2 - (R^t)^2\big)\frac{\|\boldsymbol{x}\|_2^2}{2} + \sum_{t=2}^{T}(R^t - R^{t-1})\langle\boldsymbol{x},\hat{\boldsymbol{\theta}}^t\rangle \qquad (16)$$
$$\geq\sum_{t=2}^{T}\big((R^{t-1})^2 - (R^t)^2\big)\frac{\|\boldsymbol{x}\|_2^2}{2} = \big((R^1)^2 - (R^T)^2\big)\frac{\|\boldsymbol{x}\|_2^2}{2},$$

where the first equality is from that $\mathcal{B}_\psi(\boldsymbol{a},\boldsymbol{b}) = \|\boldsymbol{a}-\boldsymbol{b}\|_2^2/2 = \|\boldsymbol{a}\|_2^2/2 - \langle\boldsymbol{a},\boldsymbol{b}\rangle + \|\boldsymbol{b}\|_2^2/2$ if $\psi(\cdot)$ is the quadratic regularizer (as stated around Eq. (3)), as well as the inequality is from the facts that $R^t \geq R^{t-1}$ (from the update rule of MI-SPRM$^+$ as shown in Eq. (6)) and $\langle\boldsymbol{x},\hat{\boldsymbol{\theta}}^t\rangle \geq 0$ (as $\boldsymbol{x} \geq \boldsymbol{0}$ and $\hat{\boldsymbol{\theta}}^t \geq \boldsymbol{0}$). From the fact that for any $R^1$, $R^t$ increases monotonically to a constant $C_3$ as $t \to \infty$ (as stated around Eq. (7)), we have $(R^T)^2 \leq C_3^2$. From Eq. (16), as well as combining $(R^T)^2 \leq C_3^2$ and $\|\boldsymbol{x}\|_2^2 = \sum_{i\in\mathcal{N}}\|\boldsymbol{x}_i\|_2^2 \leq |\mathcal{N}|$ (since $\|\boldsymbol{x}_i\|_2^2 \leq 1$ as stated around Eq. (1)), we get

$$\sum_{t=2}^{T}\Big(-\mathcal{B}_\psi(R^t\boldsymbol{x},\hat{\boldsymbol{\theta}}^t) + \mathcal{B}_\psi(R^{t-1}\boldsymbol{x},\hat{\boldsymbol{\theta}}^t)\Big) \geq \big((R^1)^2 - (R^T)^2\big)\frac{\|\boldsymbol{x}\|_2^2}{2} \geq -(R^T)^2\frac{\|\boldsymbol{x}\|_2^2}{2} \geq -C_3^2|\mathcal{N}|.$$
$$(17)$$

Then, for the term $\mathcal{B}_\psi(R^T\boldsymbol{x},\hat{\boldsymbol{\theta}}^{t+1}) - \mathcal{B}_\psi(R^1\boldsymbol{x},\hat{\boldsymbol{\theta}}^1)$, we have

$$\mathcal{B}_\psi(R^T\boldsymbol{x},\hat{\boldsymbol{\theta}}^{t+1}) - \mathcal{B}_\psi(R^1\boldsymbol{x},\hat{\boldsymbol{\theta}}^1) \geq -\mathcal{B}_\psi(R^1\boldsymbol{x},\hat{\boldsymbol{\theta}}^1). \qquad (18)$$

Let $\mathcal{B}_\psi(R^1\boldsymbol{x},\hat{\boldsymbol{\theta}}^1) \leq C_4$ with $C_4$ is a constant (such $C_4$ must exists since (i) $R^1$ and $\hat{\boldsymbol{\theta}}^1$ are given, as well as (ii) $\boldsymbol{x} \in \mathcal{X}$ with $\mathcal{X}$ is a compact set as stated around Eq.(1)). Then, combining Eq. (15), (17), and (18), we have

$$\sum_{t=1}^{T}R^t\langle\boldsymbol{\ell}^t,\boldsymbol{x}^t - \boldsymbol{x}\rangle - C_3^2|\mathcal{N}| - C_4$$
$$\leq\sum_{t=1}^{T}\big(\|\boldsymbol{F}^t(\boldsymbol{\theta}^t) - \boldsymbol{F}^{t-1}(\boldsymbol{\theta}^{t-1})\|_2^2\big) + \frac{\mathcal{B}_\psi(\hat{\boldsymbol{\theta}}^1,\boldsymbol{\theta}^0)}{2} - \sum_{t=1}^{T}\left(\frac{\mathcal{B}_\psi(\boldsymbol{\theta}^t,\hat{\boldsymbol{\theta}}^t)}{2} + \frac{\mathcal{B}_\psi(\hat{\boldsymbol{\theta}}^t,\boldsymbol{\theta}^{t-1})}{2}\right). \qquad (19)$$

To bound the value of $(\|\boldsymbol{F}^t(\boldsymbol{\theta}^t) - \boldsymbol{F}^{t-1}(\boldsymbol{\theta}^{t-1})\|_2^2 - (\mathcal{B}_\psi(\boldsymbol{\theta}^t,\hat{\boldsymbol{\theta}}^t) + \mathcal{B}_\psi(\hat{\boldsymbol{\theta}}^t,\boldsymbol{\theta}^{t-1}))/2$, we show that $(\|\boldsymbol{F}^t(\boldsymbol{\theta}^t) - \boldsymbol{F}^{t-1}(\boldsymbol{\theta}^{t-1})\|_2^2 - (\mathcal{B}_\psi(\boldsymbol{\theta}^t,\hat{\boldsymbol{\theta}}^t) + \mathcal{B}_\psi(\hat{\boldsymbol{\theta}}^t,\boldsymbol{\theta}^{t-1}))/2) > 0$ only appears

$$T_r = \lceil 2\sqrt{C_2} - R^1\rceil$$

times, where $\lceil\cdot\rceil$ is the ceiling integer of a number and $C_2$ is defined in Lemma A.3. Formally, $(\|\boldsymbol{F}^t(\boldsymbol{\theta}^t) - \boldsymbol{F}^{t-1}(\boldsymbol{\theta}^{t-1})\|_2^2 - (\mathcal{B}_\psi(\boldsymbol{\theta}^t,\hat{\boldsymbol{\theta}}^t) + \mathcal{B}_\psi(\hat{\boldsymbol{\theta}}^t,\boldsymbol{\theta}^{t-1}))/2) > 0$ implies $R^{t+1} = R^t + 1$. Also, we have that once $R^{t-1} \geq 2\sqrt{C_2}$, for any $t' \geq t-1$, $R^{t'} = R^{t-1}$ (as stated around Eq. (7)). Thus, we have that MI-SPRM$^+$ only updates the value of $R^t$ within $T_r$ times ($R^{t+1} = R^t + 1$) to ensure $R^{t-1} \geq 2\sqrt{C_2}$ since

$$R^1 + T_r = R^1 + \lceil 2\sqrt{C_2} - R^1\rceil \geq 2\sqrt{C_2}.$$

Therefore, from the facts that (i) $R^{t+1} = R^t + 1$ only appears $T_r = \lceil 2\sqrt{C_2} - R^1\rceil$ times and (ii) $R^{t+1} = R^t + 1$ appears if and only if $(\|\boldsymbol{F}^t(\boldsymbol{\theta}^t) - \boldsymbol{F}^{t-1}(\boldsymbol{\theta}^{t-1})\|_2^2 - (\mathcal{B}_\psi(\boldsymbol{\theta}^t,\hat{\boldsymbol{\theta}}^t) + \mathcal{B}_\psi(\hat{\boldsymbol{\theta}}^t,\boldsymbol{\theta}^{t-1}))/2) > 0$ (as shown in Eq. (6)), we have that $(\|\boldsymbol{F}^t(\boldsymbol{\theta}^t) - \boldsymbol{F}^{t-1}(\boldsymbol{\theta}^{t-1})\|_2^2 - (\mathcal{B}_\psi(\boldsymbol{\theta}^t,\hat{\boldsymbol{\theta}}^t) + \mathcal{B}_\psi(\hat{\boldsymbol{\theta}}^t,\boldsymbol{\theta}^{t-1}))/2) > 0$ only appears $T_r$ times. Let these $T_r$ times be denoted by the set $\mathcal{T}$, we have

$$\sum_{t=1}^{T}\left(\|\boldsymbol{F}^t(\boldsymbol{\theta}^t) - \boldsymbol{F}^{t-1}(\boldsymbol{\theta}^{t-1})\|_2^2 - \frac{\mathcal{B}_\psi(\boldsymbol{\theta}^t,\hat{\boldsymbol{\theta}}^t)}{2} - \frac{\mathcal{B}_\psi(\hat{\boldsymbol{\theta}}^t,\boldsymbol{\theta}^{t-1})}{2}\right)$$
$$\leq\sum_{t\in\mathcal{T}}\left(\|\boldsymbol{F}^t(\boldsymbol{\theta}^t) - \boldsymbol{F}^{t-1}(\boldsymbol{\theta}^{t-1})\|_2^2 - \frac{\mathcal{B}_\psi(\boldsymbol{\theta}^t,\hat{\boldsymbol{\theta}}^t)}{2} - \frac{\mathcal{B}_\psi(\hat{\boldsymbol{\theta}}^t,\boldsymbol{\theta}^{t-1})}{2}\right), \qquad (20)$$

where the inequality is from the fact that the time $t$, which is not included in $\mathcal{T}$, ensures that $(\|\boldsymbol{F}^t(\boldsymbol{\theta}^t) - \boldsymbol{F}^{t-1}(\boldsymbol{\theta}^{t-1})\|_2^2 - (\mathcal{B}_\psi(\boldsymbol{\theta}^t,\hat{\boldsymbol{\theta}}^t) + \mathcal{B}_\psi(\hat{\boldsymbol{\theta}}^t,\boldsymbol{\theta}^{t-1}))/2) \leq 0$. Continuing from Eq. (20), we

have

$$
\begin{aligned}
&\sum_{t\in\mathcal{T}}\left(\|\boldsymbol{F}^t(\boldsymbol{\theta}^t)-\boldsymbol{F}^{t-1}(\boldsymbol{\theta}^{t-1})\|_2^2-\frac{\mathcal{B}_\psi(\boldsymbol{\theta}^t,\hat{\boldsymbol{\theta}}^t)}{2}-\frac{\mathcal{B}_\psi(\hat{\boldsymbol{\theta}}^t,\boldsymbol{\theta}^{t-1})}{2}\right)\\
&\leq\sum_{t\in\mathcal{T}}\|\boldsymbol{F}^t(\boldsymbol{\theta}^t)-\boldsymbol{F}^{t-1}(\boldsymbol{\theta}^{t-1})\|_2^2\\
&\leq\sum_{t\in\mathcal{T}}\sum_{i\in\mathcal{N}}\|\langle\boldsymbol{\ell}_i^t,\boldsymbol{x}_i^t\rangle\mathbf{1}-\boldsymbol{\ell}_i^t-\langle\boldsymbol{\ell}_i^{t-1},\boldsymbol{x}_i^t\rangle\mathbf{1}+\boldsymbol{\ell}_i^{t-1}\|_2^2\\
&\leq4\sum_{t\in\mathcal{T}}\sum_{i\in\mathcal{N}}\left(\|\langle\boldsymbol{\ell}_i^t,\boldsymbol{x}_i^t\rangle\mathbf{1}\|_2^2+\|\boldsymbol{\ell}_i^t\|_2^2+\|\langle\boldsymbol{\ell}_i^{t-1},\boldsymbol{x}_i^t\rangle\mathbf{1}\|_2^2+\|\boldsymbol{\ell}_i^{t-1}\|_2^2\right)\\
&\leq4\sum_{t\in\mathcal{T}}\sum_{i\in\mathcal{N}}\left(|\mathcal{A}_i|^2\|\langle\boldsymbol{\ell}_i^t,\boldsymbol{x}_i^t\rangle\|_2^2+\|\boldsymbol{\ell}_i^t\|_2^2+|\mathcal{A}_i|^2\|\langle\boldsymbol{\ell}_i^{t-1},\boldsymbol{x}_i^{t-1}\rangle\|_2^2+\|\boldsymbol{\ell}_i^{t-1}\|_2^2\right)\\
&\leq4\sum_{t\in\mathcal{T}}\sum_{i\in\mathcal{N}}\left(D^2\|\boldsymbol{\ell}_i^t\|_2^2\|\boldsymbol{x}_i^t\|_2^2+\|\boldsymbol{\ell}_i^t\|_2^2+D^2\|\boldsymbol{\ell}_i^{t-1}\|_2^2\|\boldsymbol{x}_i^t\|_2^2+\|\boldsymbol{\ell}_i^{t-1}\|_2^2\right)\\
&\leq4\sum_{t\in\mathcal{T}}\left(D^2P^2+P^2+D^2P^2+P^2\right)=8T_rP^2(D^2+1),
\end{aligned}\tag{21}
$$

where the third inequality comes from that $\forall\boldsymbol{a},\boldsymbol{b},\boldsymbol{c},\boldsymbol{d}\in\mathbb{R}^d,\|\boldsymbol{a}+\boldsymbol{b}+\boldsymbol{c}+\boldsymbol{d}\|_2^2\leq4(\|\boldsymbol{a}\|_2^2+\|\boldsymbol{b}\|_2^2+\|\boldsymbol{c}\|_2^2+\|\boldsymbol{d}\|_2^2)$, the fourth inequality is from that $\langle\boldsymbol{\ell}_i^t,\boldsymbol{x}_i^t\rangle\mathbf{1}$ and $\langle\boldsymbol{\ell}_i^{t-1},\boldsymbol{x}_i^{t-1}\rangle\mathbf{1}$ are $|\mathcal{A}_i|$-dimensional vectors (as stated around Eq. (4)), the fifth inequality is from that $D=\max_{i\in\mathcal{N}}|\mathcal{A}_i|$, as well as the last inequality comes from the facts that $\|\boldsymbol{x}_i\|_2^2\leq\|\boldsymbol{x}_i\|_1^2=1$ (as stated around Eq.(1)) and $\sum_{i\in\mathcal{N}}\|\boldsymbol{\ell}_i\|_2^2=\|\boldsymbol{\ell}\|_2^2\leq P^2$ (Eq.(1)). Combining Eq. (19), (20), (21), and the fact that $\mathcal{B}_\psi(\hat{\boldsymbol{\theta}}^1,\boldsymbol{\theta}^0)/2\leq C_5$ ($C_5$ is a constant and must exists as $\hat{\boldsymbol{\theta}}^1$ and $\boldsymbol{\theta}^0$ are given), $\forall\boldsymbol{x}\in\mathcal{X}$ and $T\geq1$, we get

$$
\frac{\sum_{t=1}^T R^t\langle\boldsymbol{\ell}^t,\boldsymbol{x}^t-\boldsymbol{x}\rangle}{\sum_{t=1}^T R^t}\leq\frac{8T_rP^2(D^2+1)+C_3^2|\mathcal{N}|+C_4+C_5}{\sum_{t=1}^T R^t}.
$$

It finishes the proof. $\qquad\square$

## A.2 PROOF OF LEMMA A.3

*Proof.* To prove Lemma A.3, we first introduce Lemma A.4.

**Lemma A.4.** *(Proof is in Section A.3)* $\forall\boldsymbol{a},\boldsymbol{b}\in\mathbb{R}_{\geq0}^d,\|\boldsymbol{a}\|_1\geq C_1,\|\boldsymbol{b}\|_1\geq C_1,\left\|\frac{\boldsymbol{a}}{\|\boldsymbol{a}\|_1}-\frac{\boldsymbol{b}}{\|\boldsymbol{b}\|_1}\right\|_2\leq\frac{\sqrt{d}}{C_1}\|\boldsymbol{a}-\boldsymbol{b}\|_2.$

From the definition of $\|F^t(\boldsymbol{\theta}^t)-F^{t-1}(\boldsymbol{\theta}^{t-1})\|_2^2$ (as stated aroud Eq. (6)), we get

$$
\begin{aligned}
\|\boldsymbol{F}^t(\boldsymbol{\theta}^t)-\boldsymbol{F}^{t-1}(\boldsymbol{\theta}^{t-1})\|_2^2&=\sum_{i\in\mathcal{N}}\|\boldsymbol{F}_i^t(\boldsymbol{\theta}^t)-\boldsymbol{F}_i^{t-1}(\boldsymbol{\theta}^{t-1})\|_2^2\\
&=\sum_{i\in\mathcal{N}}\|\langle\boldsymbol{x}_i^t,\boldsymbol{\ell}_i^t\rangle\mathbf{1}-\boldsymbol{\ell}_i^t-\langle\boldsymbol{x}_i^{t-1},\boldsymbol{\ell}_i^{t-1}\rangle\mathbf{1}+\boldsymbol{\ell}_i^{t-1}\|_2^2.
\end{aligned}\tag{22}
$$

Continuing from the above equality, we have

$$
\begin{aligned}
&\|\boldsymbol{F}^t(\boldsymbol{\theta}^t)-\boldsymbol{F}^{t-1}(\boldsymbol{\theta}^{t-1})\|_2^2\\
&=\sum_{i\in\mathcal{N}}\|\langle\boldsymbol{x}_i^t,\boldsymbol{\ell}_i^t\rangle\mathbf{1}-\langle\boldsymbol{x}_i^{t-1},\boldsymbol{\ell}_i^{t-1}\rangle\mathbf{1}-\boldsymbol{\ell}_i^t+\boldsymbol{\ell}_i^{t-1}\|_2^2\\
&\leq\sum_{i\in\mathcal{N}}\left(2\|\langle\boldsymbol{x}_i^t,\boldsymbol{\ell}_i^t\rangle\mathbf{1}-\langle\boldsymbol{x}_i^{t-1},\boldsymbol{\ell}_i^{t-1}\rangle\mathbf{1}\|_2^2+2\|-\boldsymbol{\ell}_i^t+\boldsymbol{\ell}_i^{t-1}\|_2^2\right)\\
&=\sum_{i\in\mathcal{N}}\left(2|\mathcal{A}_i|\|\langle\boldsymbol{x}_i^t,\boldsymbol{\ell}_i^t\rangle-\langle\boldsymbol{x}_i^{t-1},\boldsymbol{\ell}_i^{t-1}\rangle\|_2^2+2\|-\boldsymbol{\ell}_i^t+\boldsymbol{\ell}_i^{t-1}\|_2^2\right)\\
&\leq\sum_{i\in\mathcal{N}}2D\|\langle\boldsymbol{x}_i^t,\boldsymbol{\ell}_i^t\rangle-\langle\boldsymbol{x}_i^{t-1},\boldsymbol{\ell}_i^{t-1}\rangle\|_2^2+2\|-\boldsymbol{\ell}^t+\boldsymbol{\ell}^{t-1}\|_2^2\\
&\leq2\sum_{i\in\mathcal{N}}D\|\langle\boldsymbol{x}_i^t,\boldsymbol{\ell}_i^t\rangle-\langle\boldsymbol{x}_i^{t-1},\boldsymbol{\ell}_i^{t-1}\rangle\|_2^2+2L^2\|\boldsymbol{x}^t-\boldsymbol{x}^{t-1}\|_2^2,
\end{aligned}\tag{23}
$$

where the third line is from the fact that $\forall \boldsymbol{a}, \boldsymbol{b} \in \mathbb{R}^d, \|\boldsymbol{a} + \boldsymbol{b}\|_2^2 \le 2\|\boldsymbol{a}\|_2^2 + 2\|\boldsymbol{b}\|_2^2$, the fourth line is from the fact that $\langle \boldsymbol{\ell}_i^t, \boldsymbol{x}_i^t \rangle \mathbf{1}$ and $\langle \boldsymbol{\ell}_i^{t-1}, \boldsymbol{x}_i^{t-1} \rangle \mathbf{1}$ are $|\mathcal{A}_i|$-dimensional vectors (as stated around Eq. (4)), the fifth line is from $D = \max_{i \in \mathcal{N}} |\mathcal{A}_i|$ (as stated around Eq. (1)), and the last inequality is from $\|\boldsymbol{\ell}^{\boldsymbol{x}} - \boldsymbol{\ell}^{\boldsymbol{x}'}\|_2 \le L\|\boldsymbol{x} - \boldsymbol{x}'\|_2$ (Eq. (1)). For the term $\sum_{i \in \mathcal{N}} D\|\langle \boldsymbol{x}_i^t, \boldsymbol{\ell}_i^t \rangle - \langle \boldsymbol{x}_i^{t-1}, \boldsymbol{\ell}_i^{t-1} \rangle\|_2^2$, we get

$$
\begin{aligned}
\sum_{i \in \mathcal{N}} D\|\langle \boldsymbol{x}_i^t, \boldsymbol{\ell}_i^t \rangle - \langle \boldsymbol{x}_i^{t-1}, \boldsymbol{\ell}_i^{t-1} \rangle\|_2^2 = & \sum_{i \in \mathcal{N}} D\|\langle \boldsymbol{x}_i^t, \boldsymbol{\ell}_i^t \rangle - \langle \boldsymbol{x}_i^{t-1}, \boldsymbol{\ell}_i^t \rangle + \langle \boldsymbol{x}_i^{t-1}, \boldsymbol{\ell}_i^t \rangle - \langle \boldsymbol{x}_i^{t-1}, \boldsymbol{\ell}_i^{t-1} \rangle\|_2^2 \\
\le & 2D \sum_{i \in \mathcal{N}} \left( \|\langle \boldsymbol{x}_i^t, \boldsymbol{\ell}_i^t \rangle - \langle \boldsymbol{x}_i^{t-1}, \boldsymbol{\ell}_i^t \rangle\|_2^2 + \|\langle \boldsymbol{x}_i^{t-1}, \boldsymbol{\ell}_i^t \rangle - \langle \boldsymbol{x}_i^{t-1}, \boldsymbol{\ell}_i^{t-1} \rangle\|_2^2 \right) \\
= & 2D \sum_{i \in \mathcal{N}} \left( \|\langle \boldsymbol{x}_i^t - \boldsymbol{x}_i^{t-1}, \boldsymbol{\ell}_i^t \rangle\|_2^2 + \|\langle \boldsymbol{x}_i^{t-1}, \boldsymbol{\ell}_i^t - \boldsymbol{\ell}_i^{t-1} \rangle\|_2^2 \right),
\end{aligned}
$$

where the first inequality comes from the fact that $\forall \boldsymbol{a}, \boldsymbol{b} \in \mathbb{R}^d, \|\boldsymbol{a} + \boldsymbol{b}\|_2^2 \le 2\|\boldsymbol{a}\|_2^2 + 2\|\boldsymbol{b}\|_2^2$. Then, we get

$$
\begin{aligned}
\sum_{i \in \mathcal{N}} D\|\langle \boldsymbol{x}_i^t, \boldsymbol{\ell}_i^t \rangle - \langle \boldsymbol{x}_i^{t-1}, \boldsymbol{\ell}_i^{t-1} \rangle\|_2^2 \le & 2D \sum_{i \in \mathcal{N}} \left( \|\boldsymbol{x}_i^t - \boldsymbol{x}_i^{t-1}\|_2^2 \|\boldsymbol{\ell}_i^t\|_2^2 + \|\boldsymbol{x}_i^{t-1}\|_2^2 \|\boldsymbol{\ell}_i^t - \boldsymbol{\ell}_i^{t-1}\|_2^2 \right) \\
\le & 2D \sum_{i \in \mathcal{N}} \left( P^2 \|\boldsymbol{x}_i^t - \boldsymbol{x}_i^{t-1}\|_2^2 + \|\boldsymbol{\ell}_i^t - \boldsymbol{\ell}_i^{t-1}\|_2^2 \right) \\
= & 2D \sum_{i \in \mathcal{N}} P^2 \|\boldsymbol{x}_i^t - \boldsymbol{x}_i^{t-1}\|_2^2 + 2D \sum_{i \in \mathcal{N}} \|\boldsymbol{\ell}_i^t - \boldsymbol{\ell}_i^{t-1}\|_2^2
\end{aligned} \tag{24}
$$

where the second inequality comes from $\|\boldsymbol{\ell}_i^t\|_2 \le \|\boldsymbol{\ell}_i^t\|_1 \le \|\boldsymbol{\ell}^t\|_1 \le P$ (Eq. (1)) with $\|\boldsymbol{x}_i^{t-1}\|_2^2 \le \|\boldsymbol{x}_i^{t-1}\|_1^2 = 1$ (as stated around Eq. (1)). Then, continuing from Eq. (24), we have

$$
\begin{aligned}
\sum_{i \in \mathcal{N}} D\|\langle \boldsymbol{x}_i^t, \boldsymbol{\ell}_i^t \rangle - \langle \boldsymbol{x}_i^{t-1}, \boldsymbol{\ell}_i^{t-1} \rangle\|_2^2 \le & 2D \sum_{i \in \mathcal{N}} P^2 \|\boldsymbol{x}_i^t - \boldsymbol{x}_i^{t-1}\|_2^2 + 2D \sum_{i \in \mathcal{N}} \|\boldsymbol{\ell}_i^t - \boldsymbol{\ell}_i^{t-1}\|_2^2 \\
\le & 2DP^2 \sum_{i \in \mathcal{N}} \|\boldsymbol{x}_i^t - \boldsymbol{x}_i^{t-1}\|_2^2 + 2D\|\boldsymbol{\ell}^t - \boldsymbol{\ell}^{t-1}\|_2^2 \\
\le & 2DP^2 \sum_{i \in \mathcal{N}} \|\boldsymbol{x}_i^t - \boldsymbol{x}_i^{t-1}\|_2^2 + 2DL^2\|\boldsymbol{x}^t - \boldsymbol{x}^{t-1}\|_2^2 \\
= & 2DP^2\|\boldsymbol{x}^t - \boldsymbol{x}^{t-1}\|_2^2 + 2DL^2\|\boldsymbol{x}^t - \boldsymbol{x}^{t-1}\|_2^2,
\end{aligned} \tag{25}
$$

where the third inequality is from $\|\boldsymbol{\ell}^{\boldsymbol{x}} - \boldsymbol{\ell}^{\boldsymbol{x}'}\|_2 \le L\|\boldsymbol{x} - \boldsymbol{x}'\|_2$ (Eq. (1)), the last equality is from $\|\boldsymbol{x}^t - \boldsymbol{x}^{t-1}\|_2^2 = \sum_{i \in \mathcal{N}} \|\boldsymbol{x}_i^t - \boldsymbol{x}_i^{t-1}\|_2^2$. Combining Eq. (22), (23), and (25), we get

$$
\begin{aligned}
\|F^t(\boldsymbol{\theta}^t) - F^{t-1}(\boldsymbol{\theta}^{t-1})\|_2^2 \le & 4DP^2\|\boldsymbol{x}^t - \boldsymbol{x}^{t-1}\|_2^2 + 4DL^2\|\boldsymbol{x}^t - \boldsymbol{x}^{t-1}\|_2^2 + 2L^2\|\boldsymbol{x}^t - \boldsymbol{x}^{t-1}\|_2^2 \\
= & (2L^2 + 4DL^2 + 4DP^2)\|\boldsymbol{x}^t - \boldsymbol{x}^{t-1}\|_2^2.
\end{aligned} \tag{26}
$$

Then, we get

$$
\begin{aligned}
(2L^2 + 4DL^2 + 4DP^2)\|\boldsymbol{x}^t - \boldsymbol{x}^{t-1}\|_2^2 = & (2L^2 + 4DL^2 + 4DP^2) \sum_{i \in \mathcal{N}} \|\boldsymbol{x}_i^t - \boldsymbol{x}_i^{t-1}\|_2^2 \\
\le & (2L^2 + 4DL^2 + 4DP^2) \frac{\sum_{i \in \mathcal{N}} |\mathcal{A}_i| \|\boldsymbol{\theta}_i^t - \boldsymbol{\theta}_i^{t-1}\|_2^2}{C_1^2} \\
\le & D(2L^2 + 4DL^2 + 4DP^2) \frac{\|\boldsymbol{\theta}^t - \boldsymbol{\theta}^{t-1}\|_2^2}{C_1^2} \\
= & D(2L^2 + 4DL^2 + 4DP^2) \frac{\|\boldsymbol{\theta}^t - \hat{\boldsymbol{\theta}}^t + \hat{\boldsymbol{\theta}}^t - \boldsymbol{\theta}^{t-1}\|_2^2}{C_1^2} \\
\le & 2D(2L^2 + 4DL^2 + 4DP^2) \frac{\|\boldsymbol{\theta}^t - \hat{\boldsymbol{\theta}}^t\|_2^2 + \|\hat{\boldsymbol{\theta}}^t - \boldsymbol{\theta}^{t-1}\|_2^2}{C_1^2} \\
= & 4D \frac{2L^2 + 4DL^2 + 4DP^2}{C_1^2} \left( \mathcal{B}_\psi(\boldsymbol{\theta}^t, \hat{\boldsymbol{\theta}}^t) + \mathcal{B}_\psi(\hat{\boldsymbol{\theta}}^t, \boldsymbol{\theta}^{t-1}) \right),
\end{aligned} \tag{27}
$$

where the first equality is from $\|\boldsymbol{x}^t - \boldsymbol{x}^{t-1}\|_2^2 = \sum_{i \in \mathcal{N}} \|\boldsymbol{x}_i^t - \boldsymbol{x}_i^{t-1}\|_2^2$, the third inequality is from the assumption that $\|\boldsymbol{\theta}_i^{t-1}\|_1$ and $\|\boldsymbol{\theta}_i^t\|_1$ are greater than $C_1$ with Lemma A.4, the third inequality

comes from the fact that $\forall \boldsymbol{a}, \boldsymbol{b} \in \mathbb{R}^d, \|\boldsymbol{a} + \boldsymbol{b}\|_2^2 \leq 2\|\boldsymbol{a}\|_2^2 + 2\|\boldsymbol{b}\|_2^2$ (in this case, $\boldsymbol{a} = \boldsymbol{\theta}^t - \hat{\boldsymbol{\theta}}^t$ and $\boldsymbol{b} = \hat{\boldsymbol{\theta}}^t - \boldsymbol{\theta}^{t-1}$), as well as the last line is from the facts that $\psi(\cdot)$ is the quadratic regularizer (as stated around Eq. (6)) and $\mathcal{B}_\psi(\boldsymbol{a}, \boldsymbol{b}) = \|\boldsymbol{a} - \boldsymbol{b}\|_2^2/2$ if $\psi(\cdot)$ is the quadratic regularizer (as stated around Eq. (3)). Therefore, combining Eq. (26) and (27), we obtain

$$\|F^t(\boldsymbol{\theta}^t) - F^{t-1}(\boldsymbol{\theta}^{t-1})\|_2^2 \leq 2\frac{2D(2L^2 + 4DL^2 + 4DP^2)}{C_1^2}\Big(\mathcal{B}_\psi(\boldsymbol{\theta}^t, \hat{\boldsymbol{\theta}}^t) + \mathcal{B}_\psi(\hat{\boldsymbol{\theta}}^t, \boldsymbol{\theta}^{t-1})\Big).$$

It completes the proof. $\qquad\square$

### A.3 PROOF OF LEMMA A.4

*Proof.* To prove Lemma A.4, we first introduce Lemma A.5.

**Lemma A.5.** *(Adapted from Proposition 1 in Farina et al. (2023))* $\forall \boldsymbol{c}, \boldsymbol{f} \in \mathbb{R}_{\geq 0}^d$, $\|\boldsymbol{c}\|_1 \geq 1$, $\|\boldsymbol{f}\|_1 \geq 1$, $\left\|\frac{\boldsymbol{c}}{\|\boldsymbol{c}\|_1} - \frac{\boldsymbol{f}}{\|\boldsymbol{f}\|_1}\right\|_2 \leq \sqrt{d}\|\boldsymbol{c} - \boldsymbol{f}\|_2$.

Now, let $\boldsymbol{a} = C_1\boldsymbol{c}$ and $\boldsymbol{b} = C_1\boldsymbol{f}$, where $\boldsymbol{c}, \boldsymbol{f} \in \mathbb{R}_{\geq 0}^d$, $\|\boldsymbol{c}\|_1 \geq 1$, and $\|\boldsymbol{f}\|_1 \geq 1$. From these conditions, we deduce that

$$\|\boldsymbol{a}\|_1 \geq C_1 \quad \text{and} \quad \|\boldsymbol{b}\|_1 \geq C_1.$$

Therefore, clearly, $\boldsymbol{a}$ and $\boldsymbol{b}$ satisfy the assumptions of Lemma A.4, i.e., $\boldsymbol{a}, \boldsymbol{b} \in \mathbb{R}_{\geq 0}^d$, $\|\boldsymbol{a}\|_1 \geq C_1$, and $\|\boldsymbol{b}\|_1 \geq C_1$. Then, we have

$$\begin{aligned}
\left\|\frac{\boldsymbol{a}}{\|\boldsymbol{a}\|_1} - \frac{\boldsymbol{b}}{\|\boldsymbol{b}\|_1}\right\|_2 &= \left\|\frac{C_1\boldsymbol{c}}{\|C_1\boldsymbol{c}\|_1} - \frac{C_1\boldsymbol{f}}{\|C_1\boldsymbol{f}\|_1}\right\|_2 \\
&= \left\|\frac{\boldsymbol{c}}{\|\boldsymbol{c}\|_1} - \frac{\boldsymbol{f}}{\|\boldsymbol{f}\|_1}\right\|_2.
\end{aligned} \tag{28}$$

By Lemma A.5, we have

$$\left\|\frac{\boldsymbol{c}}{\|\boldsymbol{c}\|_1} - \frac{\boldsymbol{f}}{\|\boldsymbol{f}\|_1}\right\|_2 \leq \sqrt{d}\|\boldsymbol{c} - \boldsymbol{f}\|_2. \tag{29}$$

Additionally, we have

$$\begin{aligned}
\|\boldsymbol{a} - \boldsymbol{b}\|_2 &= \|C_1\boldsymbol{c} - C_1\boldsymbol{f}\|_2 = C_1\|\boldsymbol{c} - \boldsymbol{f}\|_2 \\
\Leftrightarrow \|\boldsymbol{c} - \boldsymbol{f}\|_2 &= \frac{1}{C_1}\|\boldsymbol{a} - \boldsymbol{b}\|_2.
\end{aligned} \tag{30}$$

Combining Eq. (29) and (30), we obtain

$$\left\|\frac{\boldsymbol{c}}{\|\boldsymbol{c}\|_1} - \frac{\boldsymbol{f}}{\|\boldsymbol{f}\|_1}\right\|_2 \leq \frac{\sqrt{d}}{C_1}\|\boldsymbol{a} - \boldsymbol{b}\|_2. \tag{31}$$

Finally, combining Eq. (28) and (31), we obtain

$$\left\|\frac{\boldsymbol{a}}{\|\boldsymbol{a}\|_1} - \frac{\boldsymbol{b}}{\|\boldsymbol{b}\|_1}\right\|_2 \leq \frac{\sqrt{d}}{C_1}\|\boldsymbol{a} - \boldsymbol{b}\|_2.$$

This completes the proof. $\qquad\square$

# B CONVERGENCE RESULT OF SPRM$^+$

**Theorem B.1.** *SPRM$^+$ with $0 < \eta < R\sqrt{\frac{1}{8D(2L^2+4DL^2+4DP^2)}}$ ensures that the average strategy profile $\bar{\boldsymbol{x}}^T = \frac{\sum_{t=1}^{T} \boldsymbol{x}^t}{T}$ converges to an approximate NE with a rate of $O(\frac{1}{T})$.*

*Proof.* We would like to clarify that this proof is adapted from the proof of Theorem 4.2 of Farina et al. (2023). Considering the second line of Eq. (5), and using Lemma A.2 with $\boldsymbol{a} = \hat{\boldsymbol{\theta}}^t = [\hat{\boldsymbol{\theta}}_0^t; \hat{\boldsymbol{\theta}}_1^t]$, $\boldsymbol{a}' = \hat{\boldsymbol{\theta}}^{t+1} = [\hat{\boldsymbol{\theta}}_0^{t+1}; \hat{\boldsymbol{\theta}}_1^{t+1}]$, $\boldsymbol{a}^* = \boldsymbol{\theta} = [\boldsymbol{\theta}_0; \boldsymbol{\theta}_1]$ and $\boldsymbol{g} = -\boldsymbol{F}^t(\boldsymbol{\theta}^t) = [-\boldsymbol{F}_0^t(\boldsymbol{\theta}^t); -\boldsymbol{F}_1^t(\boldsymbol{\theta}^t)]$ (here, $\mathcal{A}$ is $\times_{i\in\mathcal{N}}\mathbb{R}_{\geq R}^{|\mathcal{A}_i|}$, implying $\boldsymbol{\theta} \in \times_{i\in\mathcal{N}}\mathbb{R}_{\geq R}^{|\mathcal{A}_i|}$), we have

$$\eta\langle -\boldsymbol{F}^t(\boldsymbol{\theta}^t), \hat{\boldsymbol{\theta}}^{t+1} - \boldsymbol{\theta}\rangle \leq \mathcal{B}_\psi(\boldsymbol{\theta}, \hat{\boldsymbol{\theta}}^t) - \mathcal{B}_\psi(\boldsymbol{\theta}, \hat{\boldsymbol{\theta}}^{t+1}) - \mathcal{B}_\psi(\hat{\boldsymbol{\theta}}^{t+1}, \hat{\boldsymbol{\theta}}^t). \tag{32}$$

Similarly, considering the first line of Eq. (5), and using Lemma A.2 with $\boldsymbol{a} = \hat{\boldsymbol{\theta}}^t = [\hat{\boldsymbol{\theta}}_0^t; \hat{\boldsymbol{\theta}}_1^t]$, $\boldsymbol{a}' = \boldsymbol{\theta}^t = [\boldsymbol{\theta}_0^t; \boldsymbol{\theta}_1^t]$, $\boldsymbol{a}^* = \hat{\boldsymbol{\theta}}^{t+1} = [\hat{\boldsymbol{\theta}}_0^{t+1}; \hat{\boldsymbol{\theta}}_1^{t+1}]$ and $\boldsymbol{g} = -\boldsymbol{F}^{t-1}(\boldsymbol{\theta}^{t-1}) = [-\boldsymbol{F}_0^{t-1}(\boldsymbol{\theta}^{t-1}); -\boldsymbol{F}_1^{t-1}(\boldsymbol{\theta}^{t-1})]$, we get

$$\eta\langle -\boldsymbol{F}^{t-1}(\boldsymbol{\theta}^{t-1}), \boldsymbol{\theta}^t - \hat{\boldsymbol{\theta}}^{t+1}\rangle \leq \mathcal{B}_\psi(\hat{\boldsymbol{\theta}}^{t+1}, \hat{\boldsymbol{\theta}}^t) - \mathcal{B}_\psi(\hat{\boldsymbol{\theta}}^{t+1}, \boldsymbol{\theta}^t) - \mathcal{B}_\psi(\boldsymbol{\theta}^t, \hat{\boldsymbol{\theta}}^t). \tag{33}$$

Summing up Eq. (32) and (33), and adding $\eta\langle \boldsymbol{F}^t(\boldsymbol{\theta}^t) - \boldsymbol{F}^{t-1}(\boldsymbol{\theta}^{t-1}), \hat{\boldsymbol{\theta}}^{t+1} - \boldsymbol{\theta}^t\rangle$ to both sides, we get

$$\eta\langle -\boldsymbol{F}^t(\boldsymbol{\theta}^t), \boldsymbol{\theta}^t - \boldsymbol{\theta}\rangle \leq \mathcal{B}_\psi(\boldsymbol{\theta}, \hat{\boldsymbol{\theta}}^t) - \mathcal{B}_\psi(\boldsymbol{\theta}, \hat{\boldsymbol{\theta}}^{t+1}) - \mathcal{B}_\psi(\hat{\boldsymbol{\theta}}^{t+1}, \boldsymbol{\theta}^t) - \mathcal{B}_\psi(\boldsymbol{\theta}^t, \hat{\boldsymbol{\theta}}^t)$$
$$+ \eta\langle \boldsymbol{F}^t(\boldsymbol{\theta}^t) - \boldsymbol{F}^{t-1}(\boldsymbol{\theta}^{t-1}), \hat{\boldsymbol{\theta}}^{t+1} - \boldsymbol{\theta}^t\rangle.$$

From Eq. (11), we have $\langle \boldsymbol{F}^t(\boldsymbol{\theta}^t), \boldsymbol{\theta}^t\rangle = 0$. In addition, we have $\langle \boldsymbol{F}^t(\boldsymbol{\theta}^t), R\boldsymbol{x}\rangle = \sum_{i\in\mathcal{N}}\langle\langle\boldsymbol{\ell}_i^t, \boldsymbol{x}_i^t\rangle\boldsymbol{1} - \boldsymbol{\ell}_i^t, R\boldsymbol{x}\rangle = R\langle\boldsymbol{\ell}^t, \boldsymbol{x}^t - \boldsymbol{x}\rangle$ ($\boldsymbol{x}_i \in \mathcal{X}_i$ implies $\langle\boldsymbol{1}, \boldsymbol{x}_i\rangle = 1$, as stated around Eq.(1)). Thus, by setting $\boldsymbol{\theta} = R\boldsymbol{x}$ (notably, $\boldsymbol{\theta} \in \times_{i\in\mathcal{N}}\mathbb{R}_{\geq R}^{|\mathcal{A}_i|}$), we get

$$\eta R\langle\boldsymbol{\ell}^t, \boldsymbol{x}^t - \boldsymbol{x}\rangle \leq \mathcal{B}_\psi(R\boldsymbol{x}, \hat{\boldsymbol{\theta}}^t) - \mathcal{B}_\psi(R\boldsymbol{x}, \hat{\boldsymbol{\theta}}^{t+1}) - \mathcal{B}_\psi(\hat{\boldsymbol{\theta}}^{t+1}, \boldsymbol{\theta}^t) - \mathcal{B}_\psi(\boldsymbol{\theta}^t, \hat{\boldsymbol{\theta}}^t)$$
$$+ \eta\langle \boldsymbol{F}^t(\boldsymbol{\theta}^t) - \boldsymbol{F}^{t-1}(\boldsymbol{\theta}^{t-1}), \hat{\boldsymbol{\theta}}^{t+1} - \boldsymbol{\theta}^t\rangle,$$

which implies

$$R\langle\boldsymbol{\ell}^t, \boldsymbol{x}^t-\boldsymbol{x}\rangle$$
$$\leq \frac{1}{\eta}\Big(\mathcal{B}_\psi(R\boldsymbol{x},\hat{\boldsymbol{\theta}}^t)-\mathcal{B}_\psi(R\boldsymbol{x},\hat{\boldsymbol{\theta}}^{t+1})-\mathcal{B}_\psi(\hat{\boldsymbol{\theta}}^{t+1},\boldsymbol{\theta}^t)-\mathcal{B}_\psi(\boldsymbol{\theta}^t,\hat{\boldsymbol{\theta}}^t)\Big)+\langle\boldsymbol{F}^t(\boldsymbol{\theta}^t)-\boldsymbol{F}^{t-1}(\boldsymbol{\theta}^{t-1}),\hat{\boldsymbol{\theta}}^{t+1}-\boldsymbol{\theta}^t\rangle$$
$$\leq \frac{1}{\eta}\Big(\mathcal{B}_\psi(R\boldsymbol{x},\hat{\boldsymbol{\theta}}^t)-\mathcal{B}_\psi(R\boldsymbol{x},\hat{\boldsymbol{\theta}}^{t+1})-\mathcal{B}_\psi(\hat{\boldsymbol{\theta}}^{t+1},\boldsymbol{\theta}^t)-\mathcal{B}_\psi(\boldsymbol{\theta}^t,\hat{\boldsymbol{\theta}}^t)\Big)+\|\boldsymbol{F}^t(\boldsymbol{\theta}^t)-\boldsymbol{F}^{t-1}(\boldsymbol{\theta}^{t-1})\|_2\|\hat{\boldsymbol{\theta}}^{t+1}-\boldsymbol{\theta}^t\|_2$$
$$\leq \frac{1}{\eta}\Big(\mathcal{B}_\psi(R\boldsymbol{x},\hat{\boldsymbol{\theta}}^t)-\mathcal{B}_\psi(R\boldsymbol{x},\hat{\boldsymbol{\theta}}^{t+1})-\mathcal{B}_\psi(\hat{\boldsymbol{\theta}}^{t+1},\boldsymbol{\theta}^t)-\mathcal{B}_\psi(\boldsymbol{\theta}^t,\hat{\boldsymbol{\theta}}^t)\Big)+2*\eta\frac{\|\boldsymbol{F}^t(\boldsymbol{\theta}^t)-\boldsymbol{F}^{t-1}(\boldsymbol{\theta}^{t-1})\|_2^2}{2}+\frac{\|\hat{\boldsymbol{\theta}}^{t+1}-\boldsymbol{\theta}^t\|_2^2}{2*2*\eta}$$
$$\leq \frac{1}{2\eta}\Big(\|R\boldsymbol{x}-\hat{\boldsymbol{\theta}}^t\|_2^2-\|R\boldsymbol{x}-\hat{\boldsymbol{\theta}}^{t+1}\|_2^2-\|\hat{\boldsymbol{\theta}}^{t+1}-\boldsymbol{\theta}^t\|_2^2-\|\boldsymbol{\theta}^t-\hat{\boldsymbol{\theta}}^t\|_2^2\Big)+\eta\|\boldsymbol{F}^t(\boldsymbol{\theta}^t)-\boldsymbol{F}^{t-1}(\boldsymbol{\theta}^{t-1})\|_2^2+\frac{\|\hat{\boldsymbol{\theta}}^{t+1}-\boldsymbol{\theta}^t\|_2^2}{4\eta}$$
$$\leq \frac{1}{2\eta}\Big(\|R\boldsymbol{x}-\hat{\boldsymbol{\theta}}^t\|_2^2-\|R\boldsymbol{x}-\hat{\boldsymbol{\theta}}^{t+1}\|_2^2\Big)-\frac{1}{4\eta}\Big(\|\hat{\boldsymbol{\theta}}^{t+1}-\boldsymbol{\theta}^t\|_2^2+\|\boldsymbol{\theta}^t-\hat{\boldsymbol{\theta}}^t\|_2^2\Big)+\eta\|\boldsymbol{F}^t(\boldsymbol{\theta}^t)-\boldsymbol{F}^{t-1}(\boldsymbol{\theta}^{t-1})\|_2^2,$$

where the third inequality comes from the fact that $ab \leq \rho b^2/2 + c^2/(2\rho), \forall b, c, \rho > 0$ (in this case, $b = 2\|\boldsymbol{F}^t(\boldsymbol{\theta}^t) - \boldsymbol{F}^{t-1}(\boldsymbol{\theta}^{t-1})\|_2$, $c = \|\hat{\boldsymbol{\theta}}^{t+1} - \boldsymbol{\theta}^t\|_2$, and $\rho = 2\eta$), and the penultimate inequality comes from $\mathcal{B}_\psi(\boldsymbol{a}, \boldsymbol{b}) = \|\boldsymbol{a} - \boldsymbol{b}\|_2^2/2$ ($\psi(\cdot)$ is the quadratic regularizer as stated around Eq. (5), as well as $\mathcal{B}_\psi(\boldsymbol{a}, \boldsymbol{b}) = \|\boldsymbol{a} - \boldsymbol{b}\|_2^2/2$ if $\psi(\cdot)$ is the quadratic regularizer as stated around Eq. (3)). Then,

we have

$$R\sum_{t=1}^{T}\langle\boldsymbol{\ell}^t,\boldsymbol{x}^t-\boldsymbol{x}\rangle$$

$$\leq\sum_{t=1}^{T}\frac{1}{2\eta}\left(\|R\boldsymbol{x}-\hat{\boldsymbol{\theta}}^t\|_2^2-\|R\boldsymbol{x}-\hat{\boldsymbol{\theta}}^{t+1}\|_2^2\right)+\eta\sum_{t=1}^{T}\|\boldsymbol{F}^t(\boldsymbol{\theta}^t)-\boldsymbol{F}^{t-1}(\boldsymbol{\theta}^{t-1})\|_2^2-\sum_{t=1}^{T}\frac{1}{4\eta}\left(\|\hat{\boldsymbol{\theta}}^{t+1}-\boldsymbol{\theta}^t\|_2^2+\|\boldsymbol{\theta}^t-\hat{\boldsymbol{\theta}}^t\|_2^2\right)$$

$$\leq\frac{\|R\boldsymbol{x}-\hat{\boldsymbol{\theta}}^1\|_2^2}{2\eta}+\eta\sum_{t=1}^{T}\|\boldsymbol{F}^t(\boldsymbol{\theta}^t)-\boldsymbol{F}^{t-1}(\boldsymbol{\theta}^{t-1})\|_2^2-\sum_{t=1}^{T}\frac{1}{4\eta}\left(\|\hat{\boldsymbol{\theta}}^{t+1}-\boldsymbol{\theta}^t\|_2^2+\|\boldsymbol{\theta}^t-\hat{\boldsymbol{\theta}}^t\|_2^2\right)$$

$$\leq\frac{\|R\boldsymbol{x}-\hat{\boldsymbol{\theta}}^1\|_2^2}{2\eta}+\eta\sum_{t=1}^{T}\|\boldsymbol{F}^t(\boldsymbol{\theta}^t)-\boldsymbol{F}^{t-1}(\boldsymbol{\theta}^{t-1})\|_2^2-\sum_{t=1}^{T}\frac{1}{4\eta}\left(\|\hat{\boldsymbol{\theta}}^t-\boldsymbol{\theta}^{t-1}\|_2^2+\|\boldsymbol{\theta}^t-\hat{\boldsymbol{\theta}}^t\|_2^2\right)-\frac{1}{4\eta}\|\hat{\boldsymbol{\theta}}^{T+1}-\boldsymbol{\theta}^T\|_2^2+\frac{1}{4\eta}\|\hat{\boldsymbol{\theta}}^1-\boldsymbol{\theta}^0\|_2^2$$

$$\leq\frac{\|R\boldsymbol{x}-\hat{\boldsymbol{\theta}}^1\|_2^2}{2\eta}+\eta\sum_{t=1}^{T}\|\boldsymbol{F}^t(\boldsymbol{\theta}^t)-\boldsymbol{F}^{t-1}(\boldsymbol{\theta}^{t-1})\|_2^2-\sum_{t=1}^{T}\frac{1}{4\eta}\left(\|\hat{\boldsymbol{\theta}}^t-\boldsymbol{\theta}^{t-1}\|_2^2+\|\boldsymbol{\theta}^t-\hat{\boldsymbol{\theta}}^t\|_2^2\right)+\frac{1}{4\eta}\|\hat{\boldsymbol{\theta}}^1-\boldsymbol{\theta}^0\|_2^2$$

$$\leq\frac{\|R\boldsymbol{x}-\hat{\boldsymbol{\theta}}^1\|_2^2}{2\eta}+\eta\sum_{t=1}^{T}\|\boldsymbol{F}^t(\boldsymbol{\theta}^t)-\boldsymbol{F}^{t-1}(\boldsymbol{\theta}^{t-1})\|_2^2-\sum_{t=1}^{T}\frac{1}{8\eta}\|\boldsymbol{\theta}^t-\boldsymbol{\theta}^{t-1}\|_2^2+\frac{1}{4\eta}\|\hat{\boldsymbol{\theta}}^1-\boldsymbol{\theta}^0\|_2^2,$$

where the third inequality is from $-\sum_{t=1}^{T}(\|\hat{\boldsymbol{\theta}}^{t+1}-\boldsymbol{\theta}^t\|_2^2+\|\boldsymbol{\theta}^t-\hat{\boldsymbol{\theta}}^t\|_2^2)=-\sum_{t=1}^{T}(\|\hat{\boldsymbol{\theta}}^t-\boldsymbol{\theta}^{t-1}\|_2^2+\|\boldsymbol{\theta}^t-\hat{\boldsymbol{\theta}}^t\|_2^2)-\|\hat{\boldsymbol{\theta}}^{T+1}-\boldsymbol{\theta}^T\|_2^2+\|\hat{\boldsymbol{\theta}}^1-\boldsymbol{\theta}^0\|_2^2$. Then, we have

$$R\sum_{t=1}^{T}\langle\boldsymbol{\ell}^t,\boldsymbol{x}^t-\boldsymbol{x}\rangle\leq\frac{\|R\boldsymbol{x}-\hat{\boldsymbol{\theta}}^1\|_2^2}{2\eta}+\eta\sum_{t=1}^{T}\|\boldsymbol{F}^t(\boldsymbol{\theta}^t)-\boldsymbol{F}^{t-1}(\boldsymbol{\theta}^{t-1})\|_2^2-\sum_{t=1}^{T}\frac{1}{8\eta}\|\boldsymbol{\theta}^t-\boldsymbol{\theta}^{t-1}\|_2^2+\frac{1}{4\eta}\|\hat{\boldsymbol{\theta}}^1-\boldsymbol{\theta}^0\|_2^2.$$

According to the proof of Lemma A.3 (Eq. (26) and the first three lines of Eq. (27)) with $C_1=R$ (from the update rule of SPRM$^+$, as shown in Eq. (5)), we get

$$R\sum_{t=1}^{T}\langle\boldsymbol{\ell}^t,\boldsymbol{x}^t-\boldsymbol{x}\rangle\leq\frac{\|R\boldsymbol{x}-\hat{\boldsymbol{\theta}}^1\|_2^2}{2\eta}+\eta\sum_{t=1}^{T}D(2L^2+4DL^2+4DP^2)\frac{\|\boldsymbol{\theta}^t-\boldsymbol{\theta}^{t-1}\|_2^2}{R^2}$$

$$-\sum_{t=1}^{T}\frac{1}{8\eta}\|\boldsymbol{\theta}^t-\boldsymbol{\theta}^{t-1}\|_2^2+\frac{1}{4\eta}\|\hat{\boldsymbol{\theta}}^1-\boldsymbol{\theta}^0\|_2^2.$$

According to above equation, if

$$\eta\frac{D(2L^2+4DL^2+4DP^2)}{R^2}\leq\frac{1}{8\eta}\Rightarrow\eta\leq R\sqrt{\frac{1}{8D(2L^2+4DL^2+4DP^2)}},$$

then

$$R\sum_{t=1}^{T}\langle\boldsymbol{\ell}^t,\boldsymbol{x}^t-\boldsymbol{x}\rangle\leq\frac{\|R\boldsymbol{x}-\hat{\boldsymbol{\theta}}^1\|_2^2}{2\eta}+\frac{1}{4\eta}\|\hat{\boldsymbol{\theta}}^1-\boldsymbol{\theta}^0\|_2^2,$$

which implies

$$\sum_{t=1}^{T}\langle\boldsymbol{\ell}^t,\boldsymbol{x}^t-\boldsymbol{x}\rangle\leq O(1),\ \forall\boldsymbol{x}\in\boldsymbol{\mathcal{X}}.$$

In other words, we have

$$\frac{\sum_{t=1}^{T}\langle\boldsymbol{\ell}^t,\boldsymbol{x}^t-\boldsymbol{x}\rangle}{T}\leq O(\frac{1}{T}),$$

which implies an $O(1/T)$ theoretical convergence rate. It completes the proof. $\qquad\square$

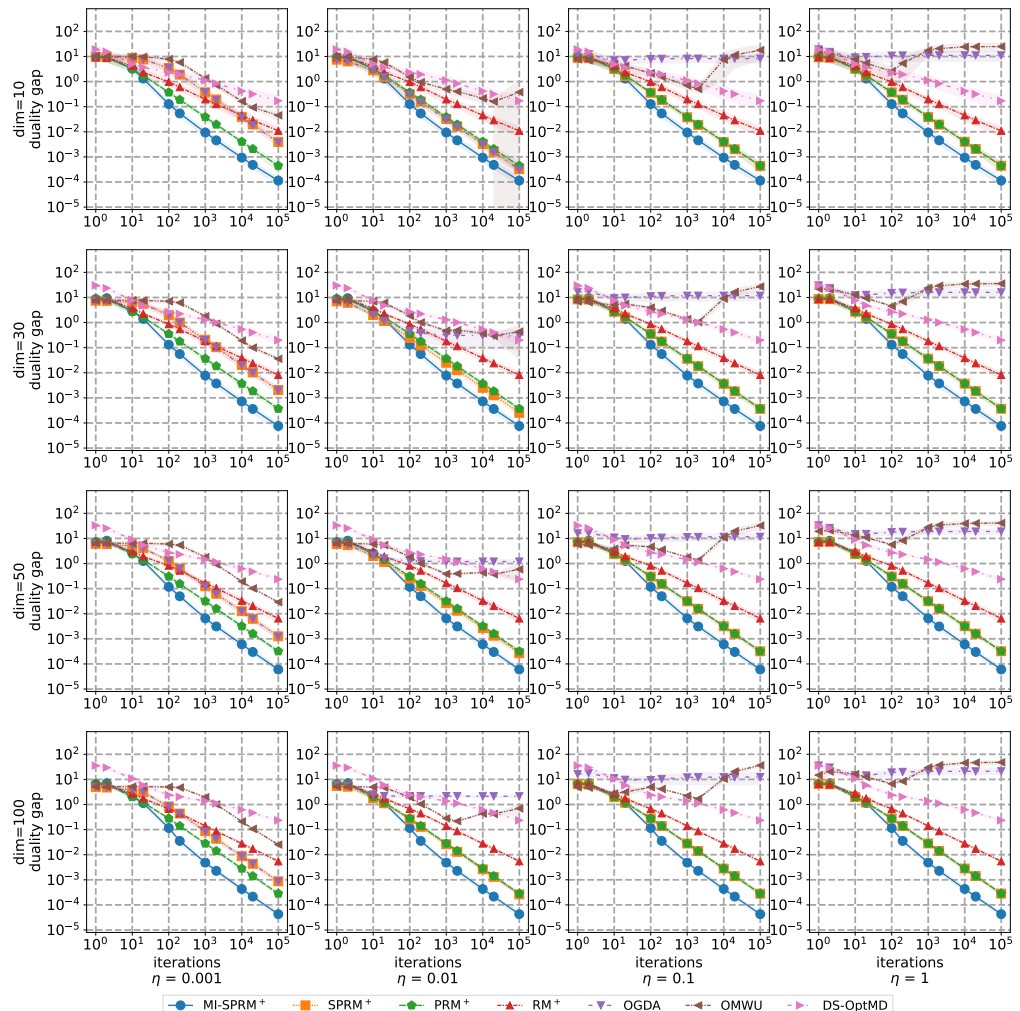

Figure 4: Convergence rates of different algorithms in randomly generated two-player zero-sum NFGs, where payoff matrices are sampled from a Gaussian distribution with mean 0 and standard deviation 10.

## C FULL EXPERIMENTAL RESULTS

**Results on convergence rates in two-player zero-sum NFGs.** We now present the experimental results omitted from Section 5. Specifically, the convergence results for randomly generated two-player zero-sum NFGs, where the payoff matrices are sampled from a Gaussian distribution with mean 0 and standard deviation 10, as well as a Gaussian distribution with mean 0 and standard deviation 1, are illustrated in Figs. 4 and 5, respectively.

**Results on convergence rates in multi-player general-sum NFGs.** Now, we evaluate the performance of MI-SPRM$^+$, RM$^+$, PRM$^+$, SPRM$^+$, OGDA, OMWU, and DS-OptMD in multi-player general-sum NFGs. Specifically, we conduct experiments on three-player general-sum NFGs of varying sizes: $[10, 30, 50]$. Due to computational constraints, we do not include results for size 100 as did in two-player zero-sum NFGs, since three-player general-sum NFGs of size 100 require computation that is 100 times greater than that for two-player zero-sum NFGs of the same size. For convenience, we generate 20 instances for each size, where the payoff matrices are drawn from a Gaussian distribution with a mean of 0 and a standard deviation of 100. Notably, similar to our experiments in two-player zero-sum NFGs, we also test a range of Gaussian distributions with varying standard deviations. However, we observe that the results do not differ significantly from those obtained with a standard deviation of 1. Consequently, this paper focuses on and reports results

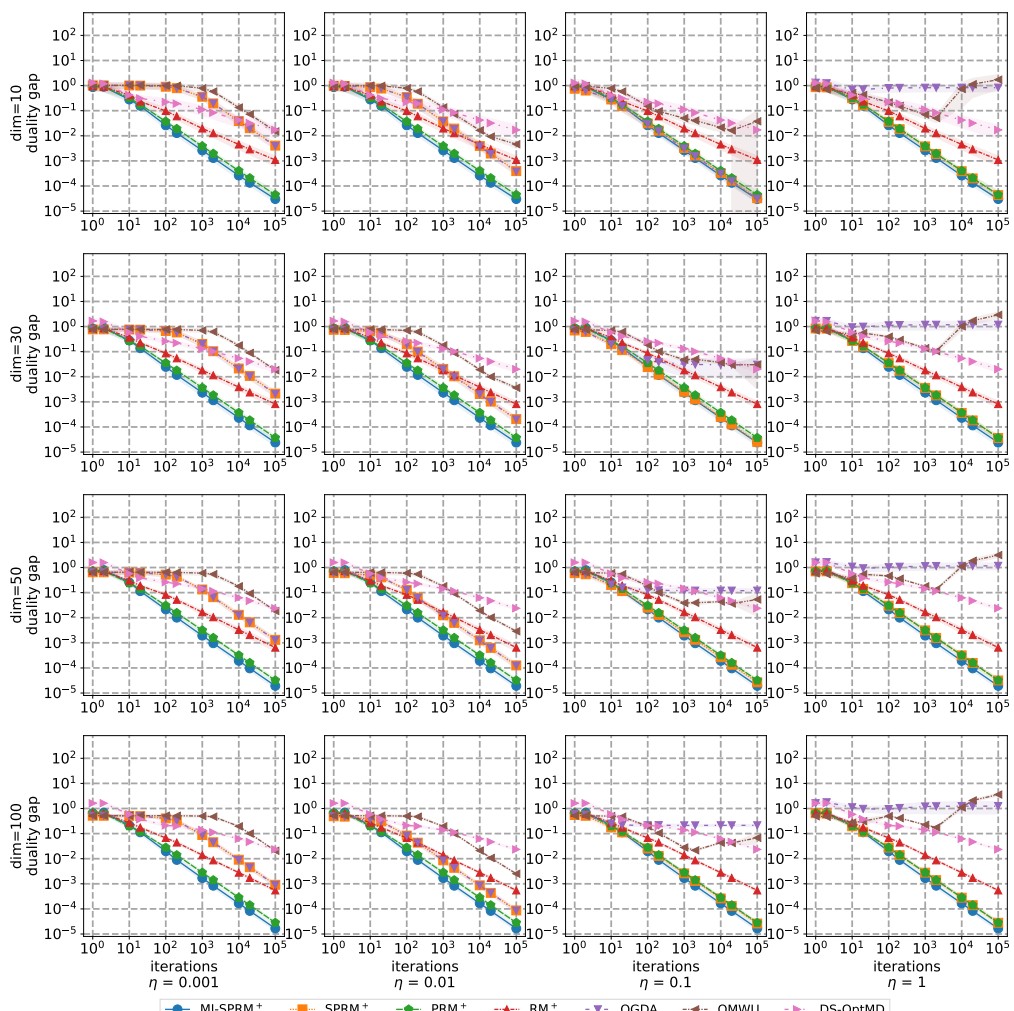

Figure 5: Convergence rates of different algorithms in randomly generated two-player zero-sum NFGs, where payoff matrices are sampled from a Gaussian distribution with mean 0 and standard deviation 1.

derived from a Gaussian distribution with a mean of 0 and a standard deviation of 1. The results are show in Fig. 6: no algorithm successfully learns an NE in all tested three-player general-sum NFGs.

**Results on dynamics of the values of $R^t$.** To validate our theoretical analysis, we examine the dynamics of $R^t$ over iterations. The results, shown in Fig. 7, align well with the theoretical analysis, showing that as the game's dimension and the standard deviation of the Gaussian sampling increase, the final value to which $R^t$ converges also increases. Specifically, theoretical analysis predicts that $R^t$ will converge near $C_2$, which is positively correlated with $L$, $P$, and $D$. As the game's dimension increases, $D$ necessarily increases. Similarly, as the standard deviation of the Gaussian sampling increases, the maximum standard value of the elements in the payoff matrix also rises, implying an increase in both $L$ and $P$. This suggests that MI-SPRM$^+$ is implicitly learning $L$ and $P$. Therefore, one of our future directions is to apply techniques for adaptively learning unknown Lipschitz constants (Malitsky and Mishchenko, 2019; Ghadimi and Lan, 2016), *i.e.*, $L$, from the field of optimization to MI-SPRM$^+$. Note that the techniques for adaptively learning unknown Lipschitz constants are related to the unconstrained strategy space while MI-SPRM$^+$ is related to the constrained strategy space. Additionally, to investigate why MI-SPRM$^+$ achieves a faster empirical convergence rate than DS-OptMD, we also present the dynamics of the step size $\eta$ in DS-OptMD[3],

---

[3]The original paper of DS-OptMD denotes the reciprocal of the step size as $\lambda_t^i$, *i.e.*, $\lambda_t^i = 1/\eta_t^i$, where $\eta_t^i$ is the step size of player $i$ at iteration $t$, we present the average step size for all players.

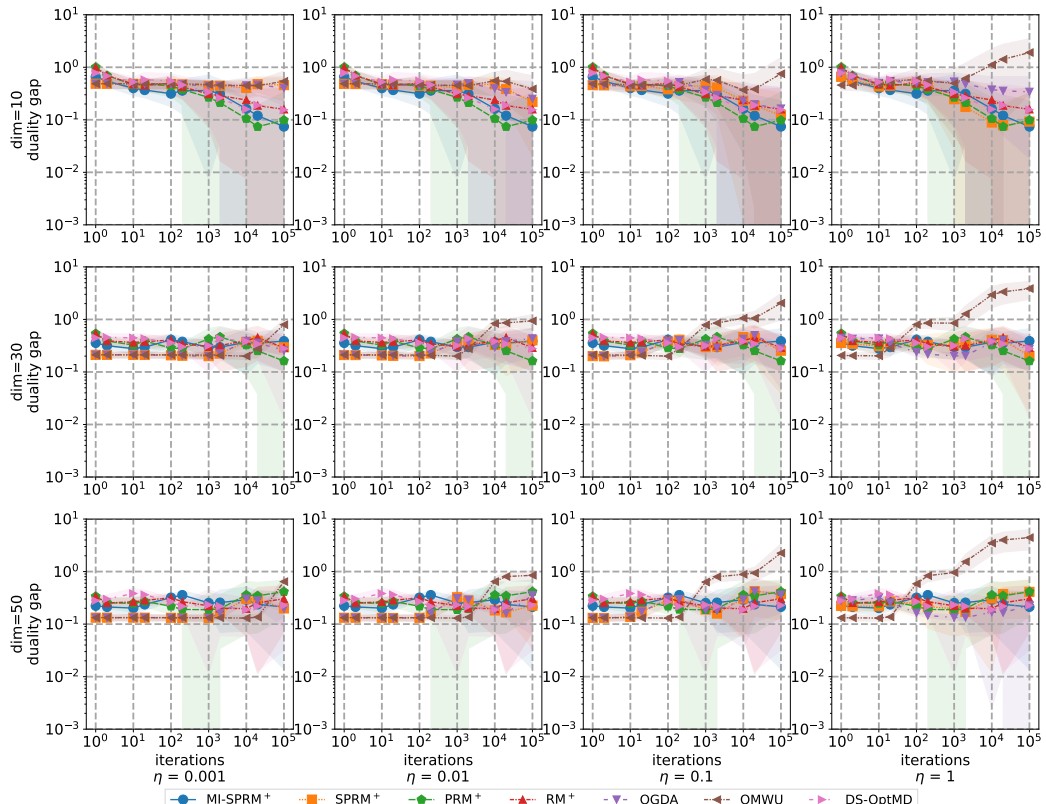

Figure 6: Convergence rates of different algorithms in randomly generated three-player general-sum NFGs, where payoff matrices are sampled from a Gaussian distribution with mean 0 and standard deviation 1.

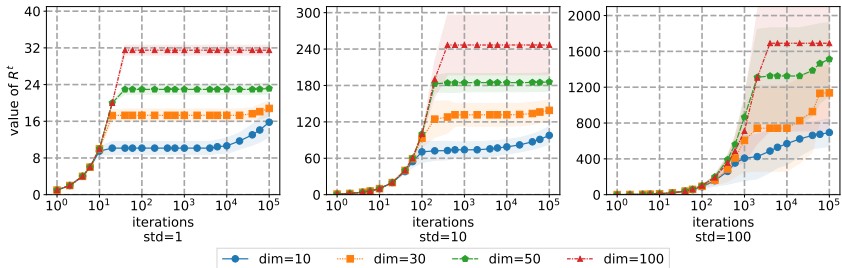

Figure 7: Dynamics of $R^t$ in two-player zero-sum NFGs. The notation "std=x" represents the standard deviation of a Gaussian distribution is x.

as shown in Fig. 8. By comparing the growth rate of $R^t$ in MI-SPRM$^+$ (Fig. 7) with the decay rate of the step size $\eta$ in DS-OptMD, it is evident that $R^t$ in MI-SPRM$^+$ grows significantly faster than the decrease in $\eta$. Therefore, we argue that the faster empirical convergence rate of MI-SPRM$^+$ compared to DS-OptMD is attributed to the step size $\eta$ in DS-OptMD. Its decay rate is too slow, requiring more iterations than MI-SPRM$^+$ to achieve the $O(1/T)$ convergence rate.

**Results on convergence rates in standard EFG benchmarks when quadratic averaging and alternating updates are used.** Now, we provide the results on convergence rates in standard EFG benchmarks when quadratic averaging and alternating updates are used. Firstly, as did in Section 5, we test on eight instances of four standard EFG benchmarks: Kuhn Poker, Leduc Poker, Liar's Dice, and Goofspiel. We compare MI-SPCFR$^+$ with SPCFR$^+$, PCFR$^+$, CFR$^+$, and DCFR. The results in Fig. 9 illustrate that, within two-player EFGs, all algorithms demonstrate improved performance when compared to scenarios where quadratic averaging and alternating updates are not used. In addition, it is important to highlight that our algorithm, MI-SPCFR+, outperforms other algorithms

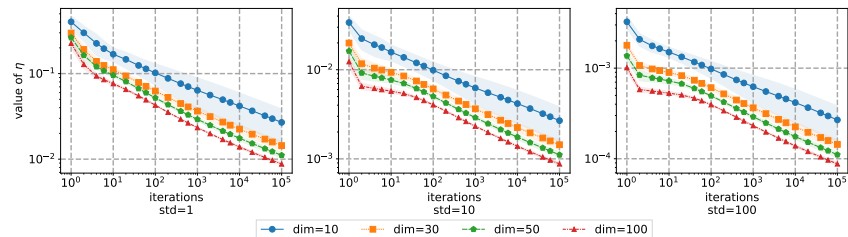

Figure 8: Dynamics of the step size $\eta$ of DS-OptMD in two-player zero-sum NFGs.

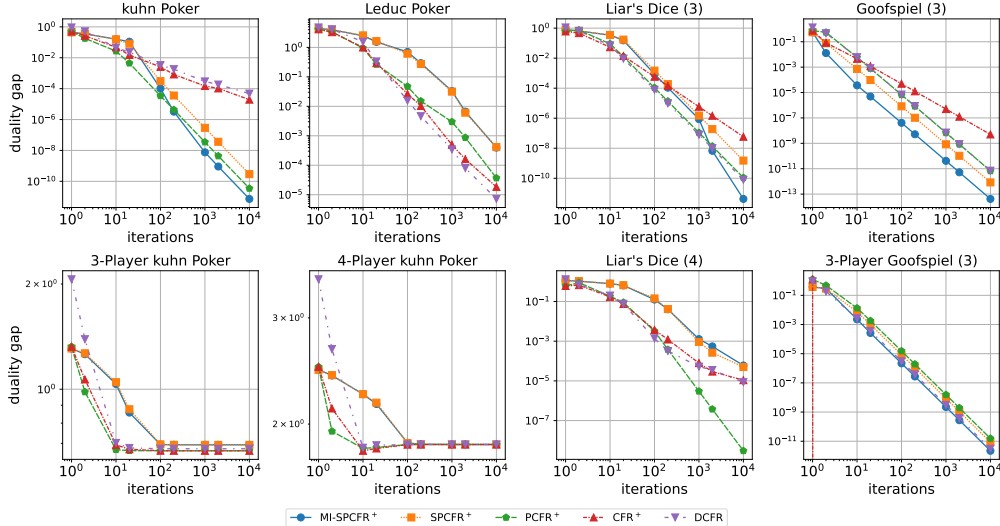

Figure 9: Convergence rates of different algorithms in standard EFG benchmarks when quadratic averaging and alternating updates are used.

Table 2: Final exploitability for the tested algorithms in HUNL Subgames. The lowest exploitability is highlighted in red.

|          | CFR$^+$ | PCFR$^+$ | SPCFR$^+$ | PDCFR$^+$ | MI-SPCFR$^+$ |
|----------|---------|----------|-----------|-----------|--------------|
| Subgame3 | 4.64e-4 | 5.14e-4  | 5.12e-4   | 4.35e-4   | 3.10e-4      |
| Subgame4 | 3.63e-4 | 4.15e-4  | 4.04e-4   | 3.89e-4   | 2.70e-4      |

in 3 of 4 tested games, such as Kuhn Poker, Leduc Poker, and Goofspiel (3). Unfortunately, in multi-player EFGs, the performance of all algorithms may significantly diminish compared to their performance without quadratic averaging and alternating updates. For example, in 3-Player Kuhn Poker and 4-Player Kuhn Poker, decreases in performance surpass a factor of 1000. In fact, in multi-player EFGs, our algorithm performs similarly to the baseline overall. No single algorithm consistently demonstrates superior performance across most games compared to others.

**Empirical convergence rates in HUNL Subgames.** To assess the performance of our MI-SPCFR$^+$ in addressing real-world games, we also conduct evaluations in HUNL Subgames, which are considerably larger than standard IIG benchmarks. Despite the presence of code related to HUNL Subgames in Openspiel, we have not successfully executed it. Therefore, we utilize HUNL Subgames implemented by Poker RL (Steinberger, 2019). Precisely, our code is based on the code from Xu et al. (2024b). The code in Xu et al. (2024b) supports only Subgame 3 and Subgame 4, so we conduct experiments solely on these two HUNL Subgames. We compare with CFR$^+$, PCFR$^+$, SPCFR$^+$, and PDCFR$^+$. We use alternating updates for each tested algorithms. Following the settings in the original version of PCFR$^+$ (Farina et al., 2021), we employ quadratic averaging for SPCFR$^+$ and MI-SPCFR$^+$. The results are shown in Table 2: MI-SPCFR$^+$ consistently outperform all baselines in both subgames.

## D    USE OF LARGE LANGUAGE MODELS

We promise that large language models are used only for editing, e.g., grammar, spelling, word choice.

