# OpenReview forum: "A Faster Parameter-Free Regret Matching Algorithm"
_ICLR.cc/2026/Conference — ICLR 2026 Poster_

### Official Review · Reviewer_yD2A · 2025-10-24

**Soundness:** 3
**Presentation:** 3
**Contribution:** 2
**Rating:** 6
**Confidence:** 3

**Summary:**

This paper introduces a smooth regret matching+ variant called Monotone Increasing Smooth Predictive Regret Matching+ (MI-SPRM+), which retains the O(1/T) convergence guarantee of current SOTA methods, while also being parameter-free. To achieve this, the authors introduce the Adaptive Regret Domain (ARD) method, which relies on the insight that the range of permissible stepsizes depends on the lower bound of the 1-norm of accumulated regrets. MI-SPRM+ dynamically adjusts the decision space after each iteration, ensuring that the lower bound monotonically increases and recovering the O(1/T) convergence property. Experiments show that MI-SPRM+ exhibits the theoretical convergence rate in standard benchmark games, and seems to outperform other existing methods in terms of duality gap.

**Strengths:**

- The paper is fairly well written, and the motivation of reducing the parameters required for tuning (especially for solving large games) is sound.
- The technical contribution and insight behind the ARD method is strong, resulting in a main theoretical result that successfully obtains strong convergence rates while being parameter free.
- The experimental results are surprisingly strong, particularly in EFGs, showing faster convergence than all prior approaches. Exploring this behavior is a crucial step for future work.

**Weaknesses:**

- The algorithm description for MI-SPRM+ is very densely written, and some more intuition would greatly improve the readability of this work.
- The paper presents a single convergence guarantee for NFGs, but it would be improved if a similar result can be shown for EFGs, even in the non-CFR variant (i.e. MI-SPRM+ directly applied to the sequence form representation of the EFG). If anything, the fact that MI-SPCFR+ seems to outperform SOTA methods makes it all the more compelling to see if there is some theoretical reasoning for this.
- While the paper shows a comparison between the compute times of the main algorithms implemented, I believe more discussion or experiments could help clarify the relationship between 1) convergence rate in terms of duality gap and 2) compute time required. In particular, it seems that MI-SPRM+ achieves parameter freeness by incurring additional adaptive steps in the process. It is however not made clear to the reader what the computational cost of parameter tuning actually is (for instance in the case of SPRM+). It seems that the parameter tuning for SPRM+ amounted to stepsizes in $\{0.01, 0.1, 1\}$ but it is not clear to me if a grid search was performed, and if that would impact the compute times accordingly.
- The sentence in Line  307: "Unfortunately, to the best of my knowledge..." is oddly phrased and could be restructured.
- "do" in Line 48 should be "are".

**Questions:**

- The authors use the phrasing "even faster than O(1/T) convergence" several times in the paper, but this is a strong statement to make, given the lack of theoretical justification. Could the faster convergence be attributed to constant factors? I would suggest changing the wording to avoid confusing readers who might misconstrue the complexity of MI-SPRM+.
- In Fig 7, we see that for some games, the value of $R^t$ seems to plateau then subsequently grow as the number of iterations increases. Is there some intuition for what this means? Would this imply that the algorithm shifts to a different NE, since the decision space changes after some period of stability?

---

> ### Author Response · Authors · 2025-11-18
>
> Thank you for your positive and encouraging feedback on our work.
>
> **W1: it would be improved if a similar result can be shown for EFGs, even in the non-CFR variant (i.e. MI-SPRM+ directly applied to the sequence form representation of the EFG).**
>
> **A:** For the result for CFR variant, as we mentioned in lines 304-312, only Clairvoyant CFR achieves an $O(1/T)$ convergence rate when learning an NE of EFGs, albeit at the cost of an $O(\log T)$ per-iteration complexity (such complexity of our MI-SPRM+ is $O(1)$).
>
> For the result for the non-CFR variant, it may lack practical value due to the significant projection overhead involved. As demonstrated by Chakrabarti et al. (2024), for a treeplex with depth $d$, number of sequences $n$, number of leaf sequences $l$, and number of infosets $m$, this overhead amounts to $O(dn \log(l + m))$. In contrast, the projection cost for the CFR variant is only $O(n)$.
>
>
>
> **W2: It is however not made clear to the reader what the computational cost of parameter tuning actually is (for instance in the case of SPRM+). It seems that the parameter tuning for SPRM+ amounted to stepsizes in but it is not clear to me if a grid search was performed, and if that would impact the compute times accordingly.**
>
> **A:** We apologize for any confusion caused. As you mentioned, parameter tuning for SPRM+ involves choosing appropriate step sizes. Notably, even without considering grid search, merely testing SPRM+ with two distinct step sizes requires more time than executing our algorithm, MI-SPRM+. This is because that as demonstrated in Table 1, MI-SPRM+'s runtime is significantly less than double that of SPRM+.
>
>
>
> **Q1: The authors use the phrasing "even faster than O(1/T) convergence" several times in the paper, but this is a strong statement to make, given the lack of theoretical justification. Could the faster convergence be attributed to constant factors?**
>
> **A:** We apologize for any confusion caused. We would like to clarify that the phrasing we use is "an $O(1/T)$ or even faster empirical convergence rate" (lines 310-311 and 430-431). Please note that we refer to the "empirical convergence rate" rather than the "theoretical convergence rate."
>
> Furthermore, the faster empirical convergence rate remains unaffected by constant factors. In our experiments, both the x-axis and y-axis are represented logarithmically. Thus, when the slope of the convergence curve is $-1$, it corresponds to an empirical convergence rate of $O(1/T)$. Importantly, constant factors do not alter the slope of the convergence curve. We observe that in many tested games, this slope can be significantly smaller. For instance, in Fig 3 depicting Liar's Dice (3), after 1e3 iterations, the slope of the convergence curve is $-3$. This indicates an empirical convergence rate of $O(1/T\^3)$, which is substantially faster than the $O(1/T)$ empirical convergence rate.
>
>
>
> **Q2: In Fig 7, we see that for some games, the value of $R\^t$ seems to plateau then subsequently grow as the number of iterations increases. Is there some intuition for what this means? Would this imply that the algorithm shifts to a different NE, since the decision space changes after some period of stability?**
>
> **A:** We hypothesize that this is because both ${\Vert F\^{t}({\theta}\^{t}) - F\^{t-1}({\theta}\^{t-1})\Vert\^2\_2}$ and $\frac{{\mathcal{B}}\_{\psi}(\hat{{\theta}}\^{t},{\theta}\^{t-1}) + {\mathcal{B}}\_{\psi}(\hat{{\theta}}\^{t},{\theta}\^{t})}{2}$ become very small as the number of iterations increase. Therefore, even a minor adjustment can lead to the condition ${\Vert F\^{t}({\theta}\^{t}) - F\^{t-1}({\theta}\^{t-1})\Vert\^2\_2} - \frac{{\mathcal{B}}\_{\psi}(\hat{{\theta}}\^{t},{\theta}\^{t-1}) + {\mathcal{B}}\_{\psi}(\hat{{\theta}}\^{t},{\theta}\^{t})}{2} > 0$ occurring, which consequently causes $R\^t$ to increase.

---

> > ### Comment · Reviewer_yD2A · 2025-11-25
> > **Response to rebuttal**
> >
> > Thank you for your response, it clarified some of my concerns with the paper. I am happy to maintain my score.

---

### Official Review · Reviewer_G2Xd · 2025-10-30

**Soundness:** 3
**Presentation:** 2
**Contribution:** 3
**Rating:** 4
**Confidence:** 5

**Summary:**

The paper proposes MI-SPRM$^+$, a smooth RM$^+$ variant that introduces an adaptive regret-domain floor so that the 1-norm of cumulative regrets grows to a level where the standard stability inequality holds. Once the floor is high enough, the scheme behaves like SPRM$^+$ without requiring tuned step sizes. The main theoretical claim is an $O(1/T)$ rate for a weighted average strategy with weights proportional to the domain floor sequence. Experiments include NFGs and EFGs in both two-player and multiplayer settings, and demonstrates favorable performance of the proposed method, when using uniform averaging for the algorithms compared to.

**Strengths:**

1. The paper studies an important problem of trying to design faster parameter-free regret-matching algorithms for equilibrium computation in games and recover parameter-free behavior while keeping the O(1/T) rate associated with smooth RM+ variants. A simple modification to SPRM+ is provided that achieves O(1/T) convergence for weighted average iterates.

2. The empirical section is broad. The method is tested in both NFGs and EFGs and compares against common RM-style and OCO baselines.

**Weaknesses:**

1. Notation and presentation need work; there is a lot of inline math in the preliminaries section which makes it difficult to read, and the notation feel excessively verbose.
2. The experiments only use uniform averaging, which is not particularly interesting from a practical perspective. It is well-established that linear or quadratic averaging should be used, or even the last iterate in some cases.
3. The experiments do not use alternation. While it is good to test on the simultaneous case to verify the theory, numerical take-aways should most likely be concluded for the alternating variant.
4. The proposed change does retain parameter-freeness, but it does not retain stepsize-invariance, another property cojectured to be important for RM-based algorithms.

**Questions:**

1. The theoretical result is based on a weighted average of the iterates, which is not standard in decentralized learning settings. While this doesn’t take away from the result, it would be good to emphasize this in the abstract and intro.

2. Why not include alternation and quadratic averaging for the CFR+/predictive CFR+ baselines, as is standard in the literature? These choices often improve empirical performance substantially on the EFGs you report.

3. In Table 1, are the reported times for a fixed number of iterations (1e5?) or for reaching a fixed duality gap?

4. Given that you run experiments for computing NE in multiplayer games (and use duality gap as the potential function), it might sense to not specify the notation/preliminaries section to use 2 players (even though the theoretical work is focused on the 2p0s setting) and let it be more general (some of the notation is already more general).

5. Why is a proof for $O(1/T)$ of SPRM$^+$ given? Isn't that just restating results from Farina et al?

---

> ### Author Response · Authors · 2025-11-18
>
> Thank you for your time and for sharing your critical perspective on our manuscript.
>
> **W1: The experiments do not use quadratic averaging and alternating updates, which often improve empirical performance substantially on the EFGs.**
>
> **A:** In the updated version of Figure 9, we include results for MI-SPCFR+, SPCFR+, PCFR+, CFR+, and DCFR, utilizing quadratic averaging and alternating updates in EFGs.
>
> Our findings indicate that, within two-player EFGs, all algorithms demonstrate improved performance when compared to scenarios where quadratic averaging and alternating updates are not used. In addition, it is important to highlight that our algorithm, MI-SPCFR+, outperforms other algorithms in 3 of 4 tested games, such as Kuhn Poker, Leduc Poker, and Goofspiel (3).
>
> Unfortunately, in multi-player EFGs, the performance of all algorithms may significantly diminish compared to their performance without quadratic averaging and alternating updates. For example, in 3-Player Kuhn Poker and 4-Player Kuhn Poker, decreases in performance surpass a factor of 1000. In fact, in multi-player EFGs, our algorithm performs similarly to the baseline overall. No single algorithm consistently demonstrates superior performance across most games compared to others.
>
>
>
> **W2: The proposed change does retain parameter-freeness, but it does not retain stepsize-invariance, another property conjectured to be important for RM-based algorithms.**
>
> **A:** We would like to clarify that the purported benefit of stepsize-invariance for RM algorithms is in question. Specifically, this stepsize-invariance is derived from the zero initialization of accumulated regret. However, recent work [1] shows that abandoning zero initialization is necessary for RM-based CFR to achieve good performance in some scenarios (Figure 3 in [1]).
>
> Additional References:
>
> [1] Linjian Meng, Tianpei Yang, Youzhi Zhang, Zhenxing Ge, Shangdong Yang, Tianyu Ding, Wenbin Li, Bo An, Yang Gao, “ Efficient Last-Iterate Convergence in Solving Extensive-Form Games”. NeurIPS 2025.
>
>
>
> **Q1: In Table 1, are the reported times for a fixed number of iterations (1e5?) or for reaching a fixed duality gap?**
>
> **A:** We apologize for any confusion caused. Table 1 reports the time for 1e5 iterations.
>
>
>
> **Q2: It might sense to not specify the notation/preliminaries section to use 2 players (even though the theoretical work is focused on the 2p0s setting) and let it be more general (some of the notation is already more general)?**
>
> **A:** The reason why we specify the preliminaries section to use 2 players is to enhance the clarity of our paper and prevent readers from mistakenly assuming that our paper focuses on learning an NE in multi-player games.
>
>
>
> **Q3: Why is a proof for of SPRM+ given? Isn't that just restating results from Farina et al?**
>
> **A:** The proof is included to ensure self-containment. The non-parameter-free convergence property of SPRM+ is a crucial motivation for our algorithm, MI-SPRM+. Moreover, proving the convergence of our algorithm is quite intricate. By offering a convergence proof for SPRM+, we aim to facilitate a better understanding of our algorithm's convergence proof and make it more accessible to the reader. In the updated version, we clearly state in Appendix B that the convergence proof for SPRM+ is adapted from Farina et al. (2023).

---

### Official Review · Reviewer_VUBu · 2025-10-31

**Soundness:** 3
**Presentation:** 3
**Contribution:** 3
**Rating:** 6
**Confidence:** 2

**Summary:**

This paper proposes a new variant of regret matching, called Monotone Increasing Smooth Predictive Regret Matching+ (MI-SPRM+), that enjoys a parameter-free setting and achieves an O(1/T) convergence rate. It differs from SPRM+ by (1) eliminating the need of using \eta, the step size of accumlating regret and (2) introduce an adaptaive scheduling of R^t, a lower bound of the regret values. The authrors also present relevant experimental results.

**Strengths:**

The technical contribution is sound.

**Weaknesses:**

The experiments lack important baselines.

**Questions:**

I think MI-SPRM+ is neat and achieving a provable O(1/T) convergence rate is great. My biggest concern is about the experiments on EFG: It would be great if baselines like CFR+ and DCFR is included.

---

> ### Author Response · Authors · 2025-11-18
>
> Thank you for your positive feedback and insightful suggestions for improvement.
>
> **W: It would be great if baselines like CFR+ and DCFR is included.**
>
> **A:** In the updated version of Figure 9, we include results for MI-SPCFR+, SPCFR+, PCFR+, CFR+, and DCFR in EFGs, utilizing quadratic averaging and alternating updates in EFGs. Our findings indicate that, within two-player EFGs, our algorithm, MI-SPCFR+, outperforms other algorithms in 3 of 4 tested games, such as Kuhn Poker, Leduc Poker, and Goofspiel (3). In multi-player EFGs, our algorithm performs similarly to the baseline overall. No single algorithm consistently demonstrates superior performance across most games compared to others.

---

> > ### Comment · Reviewer_VUBu · 2025-11-25
> >
> > Thanks for the response. Would it also be possible to test on the CFR variants of this paper: https://arxiv.org/pdf/2404.13891

---

> > > ### Author Response · Authors · 2025-11-25
> > >
> > > We have compared our MI-SPCFR+ with PDCFR+ (the algorithm you mentioned) in heads-up no-limit Texas Hold’em (HUNL) subgames, as illustrated in our responses to Reviewer CnEQ (also see the table below). Our MI-SPCFR+ outperforms PDCFR+.
> > >
> > >
> > > |           |  Subgame3   |  Subgame4   |
> > > | :-------: | :---------: | :---------: |
> > > |   CFR+    |   4.64e-4   |   3.63e-4   |
> > > |   PCFR+   |   5.14e-4   |   4.15e-4   |
> > > |  SPCFR+   |   5.12e-4   |   4.04e-4   |
> > > |  PDCFR+   |   4.35e-4   |   3.89e-4   |
> > > | MI-SPCFR+ | **3.10e-4** | **2.70e-4** |

---

### Official Review · Reviewer_CnEQ · 2025-11-01

**Soundness:** 2
**Presentation:** 3
**Contribution:** 3
**Rating:** 6
**Confidence:** 4

**Summary:**

This paper proposes Monotone Increasing Smooth Predictive Regret Matching+ (MI-SPRM+), a novel variant of regret matching (RM) that simultaneously achieves two highly desirable but previously incompatible properties in the RM family:

Parameter-free: The algorithm requires no hyperparameter tuning (e.g., step size), making it robust and practical for real-world deployment.
O(1/T) theoretical convergence rate to Nash equilibrium in two-player zero-sum normal-form games (NFGs).
To our knowledge, this is the first RM-based algorithm that attains both properties. Prior smooth RM+ methods (e.g., SPRM+, Farina et al., 2023) achieve O(1/T) convergence but lose parameter-freeness due to their dependence on a carefully tuned step size η. In contrast, classical RM+ and PRM+ are parameter-free but only guarantee O(1/√T) rates.

The key technical innovation is the Adaptive Regret Domain (ARD) mechanism, which dynamically expands the lower bound of the decision space (i.e., the 1-norm of accumulated regrets) to ensure the conditions for O(1/T) convergence are eventually satisfied—without any user-specified parameters. This approach is conceptually distinct from existing parameter-free methods like DS-OptMD, which adaptively shrink the step size and suffer from slow empirical convergence.

**Strengths:**

A rigorous theoretical analysis proving the O(1/T) convergence of the weighted average strategy under MI-SPRM+.
Comprehensive experiments on NFGs and extensive-form games (EFGs), showing that MI-SPRM+ consistently outperforms state-of-the-art baselines—including SPRM+ with optimally tuned η—in both convergence speed and final solution quality.
Empirical validation that the algorithm achieves O(1/T) (or faster) rates in practice across diverse game settings.
Overall, the work makes a clear and meaningful contribution to the regret minimization literature by resolving a concrete limitation in existing RM algorithms, with both theoretical depth and practical relevance.

**Weaknesses:**

1) Larger-scale game benchmarks: The experiments are conducted on standard small to medium-sized games (e.g., Kuhn/ Leduc Poker, random NFGs). To better demonstrate the algorithm’s scalability and practical relevance, it would be valuable to include results on larger, more challenging domains—such as a heads-up no-limit Texas Hold’em (HUNL) subgame, which is commonly used in recent game-solving literature.
2) Comparison with state-of-the-art CFR variants: The paper compares MI-SPRM+ against PRM+ and SPRM+, but does not include recently proposed advanced CFR algorithms like DDCFR[1] or PDCFR+[2]. Given that these methods also aim to accelerate convergence in EFGs, a direct comparison would help contextualize the empirical advantage of MI-SPRM+ more convincingly.

[1] Xu Hang,et al. Dynamic discounted counterfactual regret minimization.  ICLR 2024.
[2] Xu Hang,et al. Minimizing weighted counterfactual regret with optimistic online mirror descent. IJCAI 2024.

**Questions:**

See weaknesses.

---

> ### Author Response · Authors · 2025-11-18
>
> Thank you for your thoughtful comments and the positive evaluation.
>
> **W: Comparison with state-of-the-art CFR variants like DDCFR or PDCFR+ in Larger-scale game benchmarks.**
>
> **A:** We have evaluated on large-scale game benchmarks. We only report comparisons with PDCFR+ because we cannot successfully run the open-source DDCFR code. Consistent with [1] and [2], we test on Subgame 3 and Subgame 4. The results after 10,000 iterations are shown in the table below, which demonstrates that our algorithm MI-SPCFR+ outperforms CFR+, PCFR+, SPCFR+, and PDCFR+ in both two HUNL subgames. See more details in Appendix C of the updated version.
>
> |           |  Subgame3   |  Subgame4   |
> | :-------: | :---------: | :---------: |
> |   CFR+    |   4.64e-4   |   3.63e-4   |
> |   PCFR+   |   5.14e-4   |   4.15e-4   |
> |  SPCFR+   |   5.12e-4   |   4.04e-4   |
> |  PDCFR+   |   4.35e-4   |   3.89e-4   |
> | MI-SPCFR+ | **3.10e-4** | **2.70e-4** |

---

> > ### Comment · Reviewer_G2Xd · 2025-11-26
> >
> > What is the exact setup of each algorithm in the table here? Here I mean in terms of alternation, averaging, and any other parameters that need to be set.

---

> > > ### Author Response · Authors · 2025-11-26
> > >
> > > As we stated in our updated version, "We use alternating updates for each tested algorithms. Following the settings in the
> > > original version of PCFR+ (Farina et al., 2021), we employ quadratic averaging for SPCFR+ and MI-SPCFR+." For CFR+ and  PDCFR+, we follow the configurations in their original papers.

---

> > > > ### Author Response · Authors · 2025-11-26
> > > >
> > > > In CFR+, alternating updates paired with linear averaging are implemented. PCFR+, on the other hand, incorporates alternating updates alongside quadratic averaging. Similarly, SPCFR+ employs alternating updates with quadratic averaging, but additionally sets $\eta = 1$ and initializes $\\mathbf{\\theta}^1_I=\hat{\\mathbf{\\theta}}^1_I = \mathbf{1}$ for each information set $I$. In the case of PDCFR+, alternating updates are used, with the parameters set to $\alpha = 2.3$ and $\gamma = 5$. MI-SPCFR+ also employs alternating updates combined with quadratic averaging.

---

### Meta-Review · Area_Chair_za5p · 2025-12-25

**Summary:**

This work proposes a smooth regret matching variant which obtains an $O(1/T)$ rate while also being parameter-free.  The mechanism behind the algorithm is to introduce an adaptive regret-domain floor so that the 1-norm of cumulative regrets grows to a level where the standard stability inequality holds. Once the floor is high enough, the scheme behaves like an existing method without requiring tuned step sizes. Numerous experimental results are provided in the paper, which include NFGs and EFGs in both two-player and multiplayer settings. Specifically, the proposed method performs favorably compared to existing methods in the empirical results.

Overall, this work appears to be theoretically sound and makes significant contributions in Regret Matching-type algorithms, and hence is recommended an accept.

In more detail, there were some suggestions and requests from the reviewers, and the authors have taken them into account either by updating the paper or by giving appropriate replies in response to the reviewers' concerns. From my perspective, the authors have made significant efforts in responding to the reviewers' comments.

Concerns that were addressed and clarified during the rebuttal include:
- Reviewer CnEQ asked for clarification about the implementation/configuration of certain algorithms, including alternation, averaging, and other parameters that need to be set. The authors have provided sufficient details in both the rebuttal and the updated version.

- Reviewer VUBu asked whether CFR+ and DCFR could be included in the experiments, and the updated version of Figure~9 has included more baselines.

- Reviewer G2Xd suggested including alternation and quadratic averaging in the EFG experiments, and the authors have conducted EFG experiments using alternation and quadratic averaging.

- Reviewer G2Xd questioned whether the algorithm retains the parameter-free property but does not retain step size invariance. In response, the authors pointed out that a peer-reviewed related work suggests that step size invariance can lead to performance degradation, which might imply that step size invariance is not essential. The authors' response appears reasonable to me.

- Reviewer yD2A acknowledged that the authors addressed some of their concerns with the paper and indicated that they are happy to maintain their initial score of 6. The authors might want to further revise "even faster than $O(1/T)$ convergence" to "even faster than $O(1/T)$ convergence empirically" in the final version.

**Reviewer Concerns:**

Concerns that were addressed and clarified during the rebuttal include:
- Reviewer CnEQ asked for clarification about the implementation/configuration of certain algorithms, including alternation, averaging, and other parameters that need to be set. The authors have provided sufficient details in both the rebuttal and the updated version.

- Reviewer VUBu asked whether CFR+ and DCFR could be included in the experiments, and the updated version of Figure~9 has included more baselines.

- Reviewer G2Xd suggested including alternation and quadratic averaging in the EFG experiments, and the authors have conducted EFG experiments using alternation and quadratic averaging.

- Reviewer G2Xd questioned whether the algorithm retains the parameter-free property but does not retain step size invariance. In response, the authors pointed out that a peer-reviewed related work suggests that step size invariance can lead to performance degradation, which might imply that step size invariance is not essential. The authors' response appears reasonable to me.

- Reviewer yD2A acknowledged that the authors addressed some of their concerns with the paper and indicated that they are happy to maintain their initial score of 6. The authors might want to further revise "even faster than $O(1/T)$ convergence" to "even faster than $O(1/T)$ convergence empirically" in the final version.

Overall, I don't find any major concerns that are still outstanding.

**Reviewer Scores:**

The authors have made significant efforts in responding to the reviewers' comments. If the reviewers could have fully participated, they might have been more supportive or more confident in their positive ratings.

---

### Decision · Program_Chairs · 2026-01-26

Accept (Poster)